# Learning Structured Reasoning via Tractable Trajectory Control

**Po-Nien Kung** [1] **Zhen Yang** [2] **Jeffrey Luo** [1] **Cheng-Fu Yang** [1]
**Haikang Deng** [1] **Zi-Yi Dou** [2] **Yinfei Yang** [2] **Nanyun Peng** [1] **Zhe Gan** [2] **Kai-Wei Chang** [1]

## Abstract

Large language models can exhibit emergent reasoning behaviors, often manifested as recurring lexical patterns (e.g., "wait," indicating verification). However, complex reasoning trajectories remain sparse in unconstrained sampling, and standard RL often fails to guarantee the acquisition of diverse reasoning behaviors. We propose a systematic discovery and reinforcement of diverse reasoning patterns through **structured reasoning**, a paradigm that requires targeted exploration of specific reasoning patterns during the RL process. To this end, we propose Ctrl-R, a framework for learning structured reasoning via tractable trajectory control that actively guides the rollout process, incentivizing the exploration of diverse reasoning patterns that are critical for complex problem-solving. The resulting behavior policy enables accurate importance-sampling estimation, supporting unbiased on-policy optimization. We further introduce a power-scaling factor on the importance-sampling weights, allowing the policy to selectively learn from exploratory, out-of-distribution trajectories while maintaining stable optimization. Experiments demonstrate that Ctrl-R enables effective exploration and internalization of previously unattainable reasoning patterns, yielding consistent improvements across language and vision–language models on mathematical reasoning tasks.

## 1. Introduction

Language models have demonstrated strong capabilities in solving complex tasks through chain-of-thought reasoning (Wei et al., 2022; Wang & Zhou, 2024; Lyu et al., 2023). With reinforcement learning (RL), models can further improve problem-solving performance by leveraging internal

[1]University of California, Los Angeles [2]Apple. Correspondence to: Po-Nien Kung <ponienkung@ucla.edu>.

*Proceedings of the 43rd International Conference on Machine Learning*, Seoul, South Korea. PMLR 306, 2026. Copyright 2026 by the author(s).

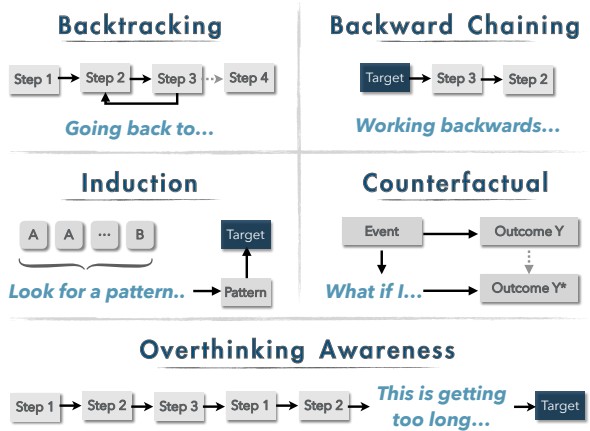

*Figure 1.* Examples of cognitive behaviors. Although cognitive behaviors are implicit, they often manifest through recurring lexical patterns during the reasoning process. We refer to reasoning that exhibits such patterns as **structured reasoning**.

reasoning processes. Prior studies show that effective solutions often rely on recurring reasoning templates, such as verification, subgoal decomposition, backtracking, and backward chaining. These structures that can be viewed as latent cognitive behaviors (Gandhi et al., 2025; Zhou et al., 2024), are closely associated with improved performance.

However, enabling models to reliably acquire and consistently utilize such reasoning structures remains challenging. Standard policy gradient methods offer limited control over which reasoning trajectories are explored and reinforced during training, particularly under sparse task rewards or at smaller model scales. As a result, useful reasoning behaviors may be under-explored, inconsistently reinforced, or fail to generalize, making it difficult to systematically study or control the learning of complex reasoning structures.

Motivated by these observations, recent approaches have explored methods for guiding language models toward desired reasoning patterns. These efforts broadly fall into three categories: supervised fine-tuning to induce distributional shifts (Deng et al., 2025), auxiliary reward design to encourage specific behaviors (Zheng et al., 2025b), and manual control over rollout trajectories during inference or training (Wang et al., 2025a). While prior approaches can bias model outputs toward certain reasoning

patterns, they lack a principled behavior policy that ensures adequate coverage of structured trajectories and supports accurate importance sampling estimation for stable policy optimization (Tokdar & Kass, 2010).

In this paper, we propose Ctrl-R, a framework for learning structured reasoning via tractable trajectory control. Ctrl-R guides RL exploration by steering the policy toward trajectories that exhibit specific reasoning structures, which are mapped to lexical patterns (see Figure 1). By defining these structures as lexical constraints over generated tokens (Table 1), we enforce them during rollout without modifying the task reward. Specifically, Ctrl-R introduces an auxiliary tractable probabilistic guidance model that estimates the likelihood of satisfying a given set of reasoning constraints during generation (Zhang et al., 2024a; 2023). This guidance model reshapes the rollout distribution, yielding an explicit, white-box behavior policy that increases exposure to trajectories exhibiting desired reasoning patterns. Crucially, the tractability of the guidance model enables accurate computation of the behavior policy probability and the corresponding importance sampling ratios during policy optimization (Tokdar & Kass, 2010). This separation enables Ctrl-R to guide reasoning exploration at rollout time without entangling heuristic constraints with the task reward, mitigating reward hacking (Fu et al., 2025).

Building on this formulation, we further introduce a power-scaling parameter for importance sampling weights that controls the strength of advantage shaping induced by guided sampling. By tuning this factor, we show that Ctrl-R can selectively amplify learning signals from exploratory, out-of-distribution trajectories while maintaining stable and conservative policy updates. Experiments in both language and vision-language model settings show that Ctrl-R consistently improves RL training on mathematical reasoning benchmarks, steering learned policies toward guided-reasoning distributions while maintaining stable optimization dynamics relative to alternative guidance strategies.

To summarize our contributions:

- We cast reasoning structures as lexical constraints to ensure their coverage during rollout exploration.
- We define a white-box behavior policy for rollout exploration, enabling accurate importance sampling estimation.
- We apply power scaling to the importance-sampling weights to selectively learn from exploratory trajectories.

**Conflict of Interest Disclosure.** The authors declare no financial conflicts of interest related to this work.

## 2. Preliminaries

**Policy Gradients and Off-Policy Learning** Standard on-policy methods, such as REINFORCE (Williams, 1992),

*Table 1.* Reasoning structures and their corresponding keyphrases, which are combined via logical OR to form lexical constraints. For example, any generation that includes *either* "working backwards" or "thinking in reverse" is considered to satisfy **Backwarding** constraint.

| *(A) Language Models* | |
| --- | --- |
| Backwarding | working backwards; thinking in reverse |
| Backtracking | let me go back; going back; undo the last step; try another way |
| Induction | try a small example; test simple numbers; look for a pattern |
| Counterfactual | what if I; imagine if I; alternatively, I could |
| Overthinking Awareness | getting too long; overcomplicating; going in circles; overthink; sufficient to answer |
| *(B) Vision–Language Models* | |
| Visual Grounding | notation; symbols; title; alignment; coordinates; layout; scale; shape; image; describe; see; identify; observe; inspect |
| General | backwards; reverse; recall; imagine; alternatively; maybe; small example; pattern |
| Reflection | wait; double-check; verify; re-examine; missed |

optimize a policy $\pi_\theta$ using trajectories $\tau$ sampled directly from the current policy. To improve sample efficiency, we can transition to off-policy learning by sampling from a behavior policy $\mu$ (typically a previous iteration of the policy) and correcting the resulting distribution mismatch via importance sampling:

$$\nabla_\theta J(\pi_\theta) = \mathbb{E}_{\tau \sim \mu} \left[ \frac{P_{\pi_\theta}(\tau)}{P_\mu(\tau)} \sum_{t=1}^{T} \nabla_\theta \log \pi_\theta(a_t \mid s_t) A_t \right],$$

where $P_{\pi_\theta}(\tau)$ and $P_\mu(\tau)$ denote trajectory likelihoods under each policy, with their ratio defining the **importance sampling weight** (Tokdar & Kass, 2010).

**PPO and the Clipped Surrogate Objective** While importance sampling is theoretically sound, large weight updates can lead to training instability. Proximal Policy Optimization (PPO) (Schulman et al., 2017) refines this framework by simplifying the global trajectory ratio into a local state-action probability ratio: $r_t(\theta) = \frac{\pi_\theta(a_t|s_t)}{\pi_{\theta_{old}}(a_t|s_t)}$. To ensure the new policy does not deviate too far from the old one, PPO employs a **clipped surrogate objective**:

$$L_{\text{PPO}}(\theta) = -\mathbb{E}_{\tau \sim \mu_\alpha} \Big[ \sum_{t=1}^{T} \min \big( r_t(\theta) A_t, \\ \text{clip}(r_t(\theta), 1 - \epsilon, 1 + \epsilon) A_t \big) \Big].$$

This clipping mechanism functions as a trust region, preserving the benefits of off-policy importance sampling while preventing unstable updates. Recent methods such as **GRPO** (Shao et al., 2024) adopt the same policy-gradient formulation but simplify advantage computation via group-relative comparisons instead of Generalized Advantage Estimation (GAE) (Schulman et al., 2015).

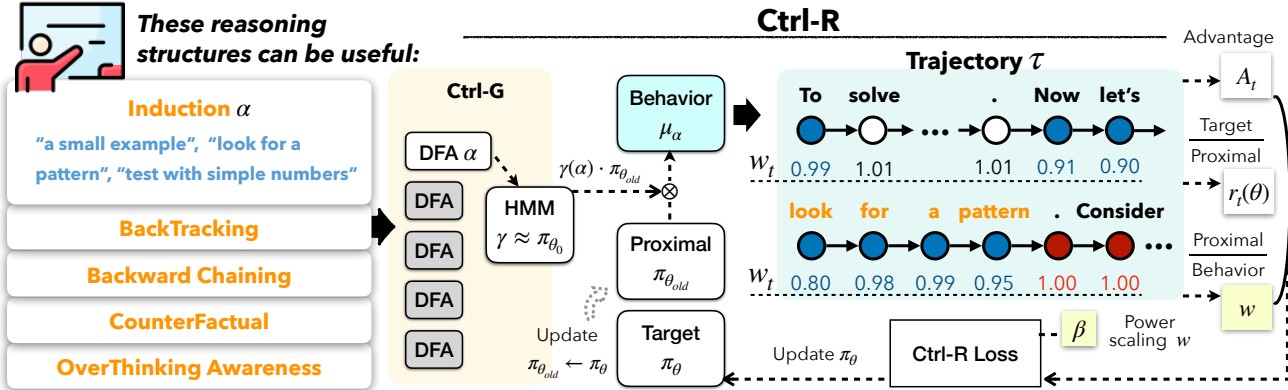

*Figure 2.* Example of sampling a guided trajectory under Ctrl-R. We first sample a constraint $\alpha$, then use the HMM to compute the marginal guidance $\gamma(\alpha \mid x_{<t}, x_t)$, which is combined with the proximal policy at each decoding step to form the guided behavior policy $\mu_\alpha$. We illustrate token-by-token decoding and visualize the guidance effect $w$ using blue, white, and red dots. Before the constraint is satisfied, sampled tokens may be dominated by either the proximal policy (white, $w > 1$) or the guidance function $\gamma$ (blue, $w < 1$). Once the constraint is satisfied, the behavior policy collapses to the proximal policy, with $w = 1$ (red).

**Problem Formulation: Learning Structured Reasoning**
Recent studies show that high-reward reasoning trajectories exhibit recurring lexical and structural patterns associated with specific cognitive behaviors, such as self-reflection and verification (Guo et al., 2025a; Gandhi et al., 2025; Wang et al., 2025b). These behaviors are often signaled by explicit tokens (e.g., "wait", "let me check"), and explicitly encouraging such structures during training has been shown to improve reinforcement learning (RL) performance (Wang et al., 2025a). Our goal is to reliably guide exploration toward these reasoning structures during RL.

To this end, we seek a behavior policy $\mu_\alpha$ that biases exploration toward trajectories exhibiting a target structure $\alpha$, while remaining compatible with importance-sampled policy optimization. Specifically, $\mu_\alpha$ should assign sufficient probability mass to structured trajectories to yield reliable learning signals, admit exact computation of $P_{\mu_\alpha}(\tau)$ for importance weighting, and remain close to the target policy $\pi_\theta$ to limit distribution mismatch and ensure stable learning.

## 3. Method

**Overview** We propose Ctrl-R, a general framework for learning structured reasoning that applies to arbitrary reasoning structures. Given a target structure $\alpha$, Ctrl-R biases rollout generation toward trajectories that exhibit $\alpha$, while enabling standard policy optimization to internalize the corresponding reasoning behavior, as illustrated in Figure 2. Ctrl-R constructs a guided behavior policy $\mu_\alpha$ within a constrained-decoding formulation by coupling the rollout policy with an explicit probabilistic guidance. This design ensures: (1) reliable exposure to the target reasoning structure during rollouts, (2) tractable importance estimation for correcting distribution mismatch, and (3) selective trajectory learning through power-scaled importance weights.

### 3.1. Structured Reasoning as Constrained Decoding

To guide language models toward trajectories $\tau = \{x_1, x_2, \ldots, x_T\}$ that elicit specific cognitive behaviors, we follow prior work (Gandhi et al., 2025; Wang et al., 2025a; Deng et al., 2025) in associating latent behaviors with observable lexical patterns $\alpha$, as summarized in Table 1. The goal is to construct a behavior policy $\mu_\alpha$ that efficiently generates trajectories satisfying these patterns within a fixed horizon $T$. Formally, we aim to maximize the probability that a sampled trajectory complies with the constraint $\alpha$:

$$\max_{\mu_\alpha} P(\tau \models \alpha \mid \mu_\alpha) = \mathbb{E}_{\tau \sim \mu_\alpha}\left[\mathbb{I}(\tau \models \alpha)\right], \quad (1)$$

where $\mathbb{I}(\cdot)$ is the indicator function.

Learning such a policy $\mu_\alpha$ directly is non-trivial. Existing approaches typically synthesize data satisfying $\alpha$ and apply supervised fine-tuning (SFT), which provides no formal guarantees of constraint satisfaction and scales poorly as each constraint requires new training data.

To address these limitations, we instead adopt inference-time constrained decoding. Rather than learning a new model to satisfy $\alpha$, we guide an existing rollout policy to comply with $\alpha$ during generation. Under an autoregressive formulation, the resulting behavior policy is:

$$\mu_\alpha(\tau = x_{1:T} \mid \alpha) = \prod_{t=1}^{T} \mu_\alpha(x_t \mid x_{<t}, \alpha). \quad (2)$$

By applying Bayes' rule and treating satisfaction of $\alpha$ as a future conditioning event, we obtain a controlled decoding policy for any base model $\pi$:

$$\mu_\alpha(x_t \mid x_{<t}, \alpha) \propto \pi(x_t \mid x_{<t}) \cdot \gamma(\alpha \mid x_t, x_{<t}), \quad (3)$$

where $\pi(x_t \mid x_{<t})$ is the model's next-token distribution, and $\gamma(\alpha \mid x_t, x_{<t})$ estimates the likelihood that the

current prefix will eventually satisfy $\alpha$. This marginal probability can be computed efficiently using probabilistic constrained decoding frameworks such as Ctrl-G (Zhang et al., 2024a) and GeLaTo (Zhang et al., 2023). More details in subsection 7.1.

### 3.2. The Ctrl-R Framework

Ctrl-R defines a *guided behavior policy* $\mu_\alpha$ that biases rollout trajectories toward a target reasoning structure $\alpha$, while preserving exact importance sampling for principled policy optimization. At each step, $\mu_\alpha$ is formed by combining the old policy $\pi_{\theta_{\text{old}}}$ with a probabilistic controller $\gamma$:

$$\mu_\alpha(x_t \mid x_{<t}) = \frac{1}{Z_t} \pi_{\theta_{\text{old}}}(x_t \mid x_{<t}) \cdot \gamma(\alpha \mid x_t, x_{<t}), \quad (4)$$

where $Z_t = \sum_{x \in \mathcal{V}} \pi_{\theta_{\text{old}}}(x \mid x_{<t}) \gamma(\alpha \mid x, x_{<t})$ normalizes the distribution. This construction yields a valid probability measure over the vocabulary, enabling stable off-policy learning with exact likelihood ratios.

**Optimization Objective**   We optimize Ctrl-R using Decoupled Proximal Policy Optimization (Hilton et al., 2022) to form the surrogate loss:

> **Ctrl-R Loss**
> $$L_{\text{Ctrl-R}}(\theta) = -\mathbb{E}_{\tau \sim \mu_\alpha} \Big[ (w)^\beta \cdot \sum_{t=1}^{T} \min \big( r_t(\theta) A_t,$$
> $$\text{clip}(r_t(\theta), 1 - \epsilon, 1 + \epsilon) A_t \big) \Big]$$

with

$$r_t(\theta) = \frac{\pi_\theta(x_t \mid x_{<t})}{\pi_{\theta_{\text{old}}}(x_t \mid x_{<t})}, \qquad w = \prod_{t=1}^{T} \frac{\pi_{\theta_{\text{old}}}(x_t \mid x_{<t})}{\mu_\alpha(x_t \mid x_{<t})}. \quad (5)$$

Here, $r_t$ denotes the token-level proximal ratio used in PPO (Schulman et al., 2017) or GRPO (Shao et al., 2024), which is clipped to enforce an implicit KL constraint (Kullback & Leibler, 1951; Schulman et al., 2017), while $w$ is the trajectory-level importance-sampling weight, which we further modulate using a power-scaling factor $\beta$.

**Design Choices**   Ctrl-R is guided by three design choices motivated by the learning dynamics of guided exploration.

**Tractable Guidance**   Rather than enforcing hard constraints, Ctrl-R introduces a tractable probabilistic guide $\gamma$ composed with the rollout policy. This yields a whitebox behavior policy that (i) ensures coverage of constrained trajectories, (ii) admits exact importance sampling via a white-box likelihood, and (iii) remains close to the target policy by construction.

**Decouple Proximal Policy Optimization**   Guided rollouts are sampled from an augmented behavior policy $\mu_\alpha$, which differs from the proximal policy assumed in standard PPO. Ctrl-R therefore adopts a decoupled formulation (Hilton et al., 2022), separating the proximal policy that controls policy updates from the behavior policy used for off-policy correction. This decoupling also makes the role of guidance explicit. Crucially, if the importance sampling correction is ignored, the resulting gradient becomes

$$\nabla \tilde{J}(\theta) = \mathbb{E}_{x \sim \mu_\alpha} \big[ \nabla_\theta \log \pi_\theta(x) A(x) \big] \quad (6)$$

$$= \frac{1}{Z} \mathbb{E}_{x \sim \pi_{\theta_{\text{old}}}} \big[ \nabla_\theta \log \pi_\theta(x) \gamma(x) A(x) \big], \quad (7)$$

which corresponds to optimizing an *advantage shaped by the control strength*. This shows that, without importance sampling correction, guided sampling not only alters exploration but also directly reshapes the advantage signal used for optimization.

**Power-Scaled Importance Weights**   Motivated by the observation in Equation 7, we propose to control the strength of this shaping effect by applying an explicit power-scaling factor $\beta \in [0, 1]$ to the importance sampling weight.

This induces a smooth tradeoff between objective-faithful optimization and control-shaped learning:

- $\beta = 1$: exact importance sampling correction; guidance affects exploration only;

- $\beta = 0$: no correction; learning is fully shaped at the advantage level by the controller;

- $0 < \beta < 1$: partial correction, yielding tempered advantage shaping with reduced variance.

As we will show empirically in section 6, intermediate values of $\beta$ amplify the learning signal from out-of-distribution trajectories, allowing the policy to selectively learn from exploratory experiences.

### 3.3. Ctrl-R Implementation

We present the instantiation of Ctrl-R with the recent constrained decoding framework *Ctrl-G* (Zhang et al., 2024a).

**Preparation: Hidden Markov Model Distillation**   To efficiently compute the marginal probability $\gamma(\alpha \mid x_{<t}, x_t) \approx p_{LM}(\alpha \mid x_{<t}, x_t)$, we follow GeLaTo (Zhang et al., 2023) and Ctrl-G (Zhang et al., 2024a) to distill a Hidden Markov Model (HMM) that approximates the initial target policy, $\gamma(x_{1:n}) \approx \pi_{\theta_0}(x_{1:n})$. We then follow the Ctrl-G design to marginalize the HMM over lexical constraints represented as a Deterministic Finite Automaton (DFA), enabling efficient computation of $\gamma(\alpha \mid x_{<t}, x_t)$. This distillation is performed once and reused for all constraints during RL.

**Preparation: Identifying Reasoning Structures** To guide RL exploration, we first specify target reasoning heuristics, which may be identified by humans or LLMs. Following prior work (Gandhi et al., 2025; Deng et al., 2025; Zhou et al., 2024), we associate cognitive behaviors with lexical patterns (Table 1) and encode them as DFAs.

**Training with Ctrl-R** We illustrate the Ctrl-R learning pipeline in Figure 2. Given a set of predefined reasoning structures with constructed DFAs, we sample one structure for each trajectory exploration and use it as a logical constraint, which is joined with a tractable HMM to compute the guiding function $\gamma(\alpha \mid x_{<t}, x_t)$ at each token generation step. This guiding model is then combined with the proximal policy $\pi_{\theta_{\text{old}}}$, which is initialized from the target policy at the start of each global batch, to form the behavior policy

$$\mu_{\alpha}(x_t \mid x_{<t}) = \frac{1}{Z_t} \pi_{\theta_{\text{old}}}(x_t \mid x_{<t}) \cdot \gamma(\alpha \mid x_{<t}, x_t).$$

We then sample a trajectory $\tau$ from this behavior policy. To correctly account for the trajectory-level importance weight $w = \prod_{t=1}^{T} w_t$, we compute the per-token weight

$$w_t = \frac{\pi_{\theta_{\text{old}}}(x_t \mid x_{<t})}{\mu_{\alpha}(x_t \mid x_{<t})} = \frac{\pi_{\theta_{\text{old}}}(x_t \mid x_{<t})}{\pi_{\theta_{\text{old}}}(x_t \mid x_{<t})} \cdot \frac{Z_t}{\gamma(\alpha \mid x_{<t}, x_t)}.$$

A smaller $w_t$ indicates that the token is strongly influenced by the HMM-based guidance, whereas a larger $w_t$ indicates that the token is dominated by the proximal policy. We illustrate the per-token dynamics in Figure 2. When early tokens are sampled from $\mu_{\alpha}$ (indicated by blue and white dots), some tokens are more favored by the proximal policy (white, $w_t > 1$), whereas others are strongly guided by the guiding function $\gamma$ (blue, $w_t < 1$). Once the constraint is satisfied at token $k - 1$, the guiding function becomes uniform over the vocabulary, causing the behavior policy to collapse to the proximal policy

$$\mu_{\alpha}(x_t \mid x_{<t}) = \pi_{\theta_{\text{old}}}(x_t \mid x_{<t}), \quad \text{for } t \geq k.$$

As a result, all subsequent tokens have $w_t = 1.0$, since no additional guidance is applied. After generating trajectories, we compute rewards and derive advantages using the Group-Related Advantage (Shao et al., 2024). We then compute the Ctrl-R loss (Equation 3.2) with an additional power-scaling factor $\beta$ to update the target policy $\pi_{\theta}$. After several gradient updates, the proximal policy is synchronized with the target policy to initiate the next training cycle.

**Implementation Details** We integrate the Ctrl-G (Zhang et al., 2023) codebase on the recent efficient LM inference package vLLM (Kwon et al., 2023), and implement Ctrl-R on RL training packages verl (Sheng et al., 2024) and EasyR1 (Zheng et al., 2025a), to enable efficient training

with Ctrl-R on both language models and vision-language models. Code is available at https://github.com/billkunghappy/Ctrl-R.

# 4. Experiments

To verify the effectiveness of Ctrl-R, we evaluate it on mathematical-reasoning benchmarks in both language-model (LM) and vision-language-model (VLM) settings.

**Mathematical Reasoning in Language Models** We evaluate Ctrl-R on mathematical reasoning tasks under the Dynamic sAmpling Policy Optimization (DAPO) setting (Yu et al., 2025). We use Qwen3-1.7B-Base and Qwen3-8B-Base (Yang et al., 2025) as base models and train them on the DAPO-Math-17K dataset with the GRPO objective (Shao et al., 2024) for 960 steps to form the base models. We then continue RL training with Ctrl-R and baseline methods from steps 960–1600. For evaluation, we follow prior work and report performance on AIME'24 (MAA, 2024), AIME'25, MATH500 (Hendrycks et al., 2021), AMC'23, Minerva (Lewkowycz et al., 2022), and Olympiad (He et al., 2024), taking the highest achievable performance across training checkpoints.

**Mathematical Reasoning in Vision–Language Models** We further study Ctrl-R in the vision–language setting, following the reinforcement learning setup of OpenVL-Thinker (Deng et al., 2025). We use Qwen2.5-VL-7B-Instruct (Bai et al., 2025) as the base model and train it on the OpenVLThinker-grpo-medium dataset with GRPO to form the base model. We then continue training on the OpenVLThinker-grpo-hard dataset for Ctrl-R and all baseline methods. For evaluation, we follow OpenVL-Thinker (Deng et al., 2025) and report scores on Math-Vista (Lu et al., 2023), MathVision (Wang et al., 2024), MathVerse (Zhang et al., 2024b), MMMU-Pro (Yue et al., 2024), Emma (Hao et al., 2025), and HallusionBench (Guan et al., 2024). For additional implementation details, including HMM distillation, see Appendix A.

**Baselines** We compare Ctrl-R with other guided policy methods for reinforcement learning on language models.

**No Guidance.** Standard RL training following our experiment setting. For LM, we use the DAPO (Yu et al., 2025) setting. For VLM, we train the Qwen2.5-VL-7B with GRPO.

**Natural Language (NL) Guidance.** Many recent methods for guiding reinforcement learning on language models apply natural language guidance by providing verbalized hints in the instruction or the response to guide the trajectory exploration (Wang et al., 2025a; Guo et al., 2025b; Nath et al., 2025). For fair comparison, we construct verbalized hints for reasoning behaviors, listed in Table 1. During the

*Table 2.* Performance comparison on math reasoning benchmarks for language models. We report Acc@16 or Acc@1 for all benchmarks. $\Delta$ denotes the absolute difference in average performance relative to the DAPO baseline. $\beta$ is the power scaling factor in Equation 3.2.

| Method | AIME'24 | AIME'25 | MATH500 | AMC'23 | Minerva | Olympiad | Avg. | |
|---|---|---|---|---|---|---|---|---|
| | Acc@16 | Acc@16 | Acc@1 | Acc@16 | Acc@1 | Acc@1 | Score | $\Delta$ |
| *Qwen3-8B-Base* | | | | | | | | |
| DAPO | 36.04 | 30.41 | 88.40 | 80.62 | **36.76** | 57.12 | 54.89 | – |
| NL Guidance | 34.58 | 31.25 | **89.40** | 81.25 | 36.39 | 57.27 | 55.02 | +0.13 |
| Reward Shaping | 34.79 | **31.87** | 89.20 | 81.87 | 34.92 | **59.05** | 55.28 | +0.39 |
| Ctrl-R ($\beta = 0.2$) | **40.00** | 31.25 | 89.20 | **82.18** | 36.39 | 58.60 | **56.27** | **+1.38** |
| *Qwen3-1.7B-Base* | | | | | | | | |
| DAPO | 13.12 | 10.21 | 68.40 | 47.81 | 23.16 | 32.78 | 32.58 | – |
| NL Guidance | 14.16 | 10.00 | 69.00 | 47.65 | **23.52** | **33.82** | 33.02 | +0.44 |
| Reward Shaping | 14.37 | 10.00 | 69.00 | 47.96 | 23.16 | 32.78 | 32.88 | +0.30 |
| Ctrl-R ($\beta = 0.2$) | **15.21** | **10.83** | **71.00** | **49.69** | 23.16 | 33.23 | **33.85** | **+1.27** |
| $\beta = 0$, *No IS* | 12.08 | 9.16 | 69.60 | 45.46 | 22.42 | 32.04 | 31.79 | **-0.79** |
| $\beta = 1$, *Full IS* | 12.29 | 9.16 | 69.6 | 46.71 | **24.26** | 32.78 | 32.46 | -0.12 |

*Table 3.* Performance comparison on math reasoning benchmarks for vision-language models. We report Acc@1 for all benchmarks. $\Delta$ denotes the absolute difference in average performance relative to the GRPO baseline. $\beta$ is the power scaling factor in Equation 3.2

| Method | MathVista | MathVerse | MathVision | MMMU-Pro | EMMA | Hallubench | Avg. | $\Delta$ |
|---|---|---|---|---|---|---|---|---|
| GRPO | 68.10 | 40.61 | 25.39 | 42.66 | 25.62 | 65.09 | 44.57 | – |
| Trajectory-Hint | 69.80 | **43.22** | 27.43 | 44.34 | 28.05 | 66.14 | 46.49 | +1.92 |
| Auxiliary Reward | **71.20** | 42.92 | **28.06** | 44.18 | 28.94 | 60.78 | 46.01 | +1.44 |
| OpenVLThinker | **71.20** | 41.42 | 25.79 | 43.06 | 29.68 | **68.45** | 46.60 | +2.03 |
| Ctrl-R ($\beta = 0.2$) | 70.90 | 42.72 | 27.14 | **44.72** | **29.93** | 67.62 | **47.17** | **+2.60** |

trajectory generation, we randomly inject these NL hints into the prompts or the start of the response.

**Reward Shaping.** A common baseline for guiding exploration is to augment the task reward $R$ with a binary shaping function $F$ (Ng et al., 1999; Guo et al., 2025a; Zheng et al., 2025b). The shaping function incentivizes the model to exhibit specific reasoning structures listed in Table 1. To mitigate reward hacking, $F$ is activated only when the final answer is correct. The resulting reward $R'$ is defined as:

$$R' = R + F, \tag{8}$$

where $R \in \{-1, 1\}$ denotes ground-truth correctness and

$$F = \begin{cases} 1, & \text{if } R > 0 \text{ and the target pattern matches,} \\ 0, & \text{otherwise.} \end{cases} \tag{9}$$

**SFT Pre-alignment.** Prior work shows that small models can acquire reasoning behaviors via supervised fine-tuning (SFT) on chain-of-thought demonstrations generated by larger models (Wei et al., 2022; Kim et al., 2023; Li et al., 2023). To bias models toward explicit reasoning structures, OpenVLThinker (Deng et al., 2025) synthesizes data with denser transitional reasoning keyphrases from stronger LMs and performs iterative SFT–RL. The SFT stage serves as a pre-alignment step that encourages the model to explore and

emit these keyphrases, while the subsequent RL stage learns how to use them correctly to improve task performance. Since this method leverages stronger reasoning models, we consider it an additional strong baseline in our VLM setting.

## 5. Results

We demonstrate the performance of reinforcement learning through guided exploration using Ctrl-R and analyze how reasoning structures are utilized during both exploration and evaluation phases.

### 5.1. Benchmark Results

**Math Reasoning – Language Models** Table 2 reports the math reasoning performance of 1.7B and 8B language models. Ctrl-R consistently improves average performance across benchmarks, achieving gains of $+1.38\%$ and $+1.27\%$ over the DAPO baseline at the 1.7B and 8B scales, respectively. These improvements are largely consistent across benchmarks; while Ctrl-R does not improve performance on Minerva, it does not introduce notable degradation.

In contrast, the *NL Guidance* and *Reward Shaping* baselines exhibit weaker and less consistent gains. Although both methods improve AIME'24 performance at the 1.7B scale,

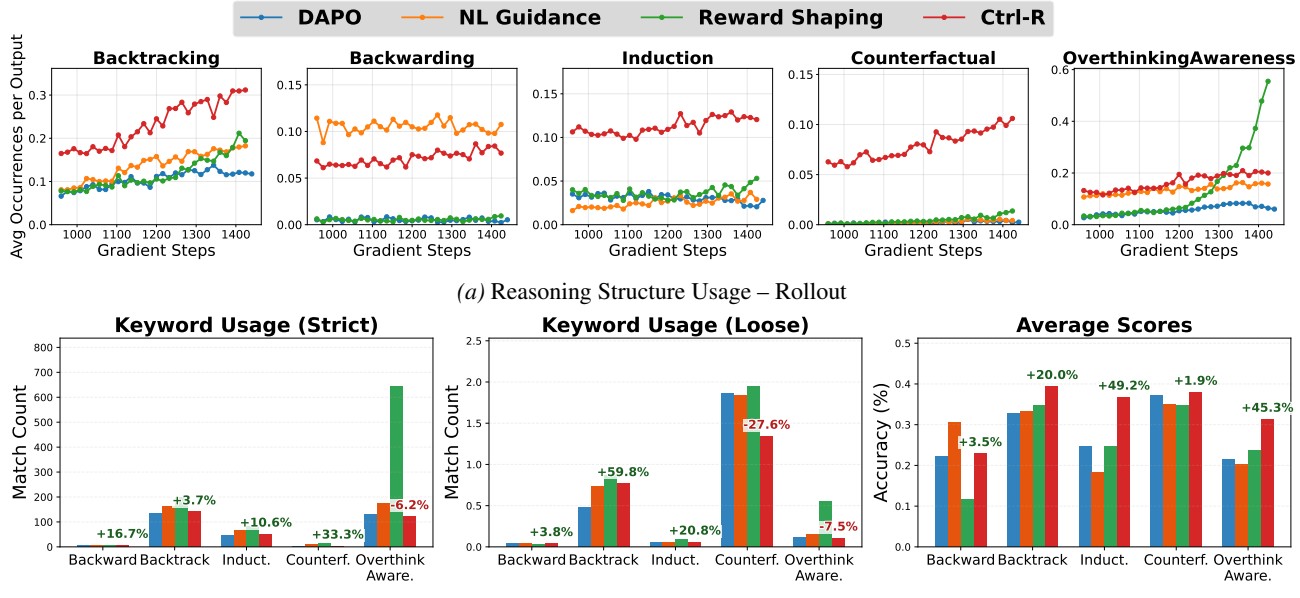

*(a)* Reasoning Structure Usage – Rollout

*(b)* Reasoning Structure Analysis – Math500 Evaluation

*Figure 3.* Trends in reasoning module usage during training and evaluation. The top panel shows rollout-time behavior. Keyword Usage (Strict) measures strict regex matches of the enforced keyphrases defined in Table 1. Keyword Usage (Loose) captures lexical variants via expanded regex patterns (Table 5). Average Scores indicate the accuracy of outputs exhibiting the corresponding reasoning patterns, as identified by loose keyword matches. All figures highlight the relative percentage change of Ctrl-R compared to the DAPO setting.

these gains do not persist when scaling to 8B models and often degrade performance on other benchmarks. Overall, Ctrl-R outperforms all baselines at both model scales, indicating that explicitly guiding training toward structured reasoning patterns yields more reliable improvements.

**Math Reasoning – Vision-Language Models**   Table 3 presents the results on math reasoning benchmarks for vision-language models. While all trajectory-guided baselines demonstrate improvements over the GRPO baseline, Ctrl-R achieves the most substantial gains, particularly on **MMMU** and **EMMA**. Notably, Ctrl-R also surpasses the **OpenVLThinker** pre-alignment baseline, a setting that requires multi-stage supervised fine-tuning and the synthesis of additional data from stronger reasoning models.

Results across both language and vision-language models demonstrate that reinforcement learning with guided exploration via Ctrl-R effectively enhances reasoning performance across diverse settings.

### 5.2. Learning Dynamics of Reasoning Structures

**Reasoning structure coverage during rollout**   To understand how Ctrl-R enhances mathematical reasoning, we analyze the evolution of reasoning structures explored during training. Figure 3a illustrates the frequency and distribution of these structures during rollout exploration.

We observe that by integrating an auxiliary Hidden Markov

Model (HMM) to guide trajectory exploration, the Ctrl-R effectively steers the rollout policy toward generating target reasoning patterns. Notably, this approach achieves sufficient coverage of complex structures that remain unobserved in the baseline model. This ensures that the target policy is trained on trajectories exhibiting the desired lexical patterns, allowing it to internalize the corresponding reasoning behavior.

In contrast, *NL Guidance* attempts to bias exploration through natural-language instructions, but the model does not reliably follow these directives. As a result, usage of structures such as **Induction** and **Counterfactual** remains low. For *Reward Shaping*, reasoning structures are not explicitly encouraged during rollout. Consequently, underexplored behaviors such as **Backwarding** and **Counterfactual** remain rare throughout training, as the shaping reward is seldom activated. Conversely, structures like **Overthinking Awareness** are overly reinforced, leading to disproportionately high usage.

**Internalization of reasoning structures**   To assess how the target policy internalizes reasoning behaviors, we analyze reasoning structure usage in the evaluation outputs of **Math500** benchmark and report the results in Figure 3b. While *Reward Shaping* substantially increases the usage of **Overthinking Awareness**—particularly for directly rewarded keyphrases (strict keywords), this increase does not translate into meaningful improvements in average task per-

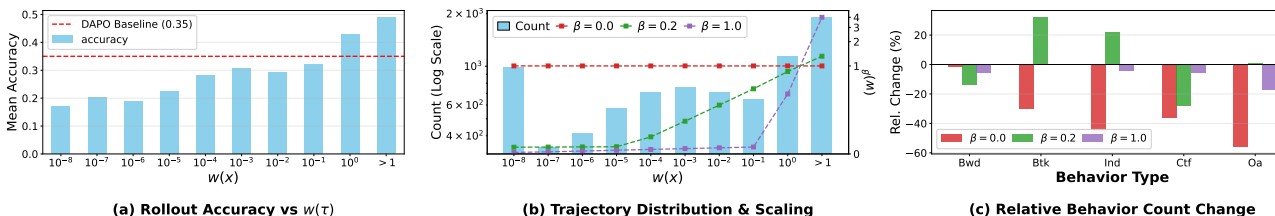

*(a)* **Rollout Accuracy vs.** $w(\tau)$: The dashed red line denotes overall accuracy. **Trajectory Distribution and Scaling**: Histogram of trajectory counts across weight bins, with overlaid curves showing the effect of $(w)^\beta$ on the loss; values near zero contribute little learning signal. **Relative Behavior Change**: Percentage change in reasoning behaviors across models trained with different $\beta$.

| **Noisy** $w < 10^{-6}$ | **Exploratory** — $10^{-6} < w < 10^{-1}$ | **On-Policy** $w > 10^{-1}$ |
|---|---|---|
| "... triangle is:\n\n\\[\\boxed{3}\\] ... \n\n### ŏ ow rifting\n!我要我\n\n##### ### ### ### Going back to the soldiers whose chests..." | "... Wait, perhaps a better approach. Let's look for a pattern (forced pattern) or a common base.\n\nRecall that..." | "... perpendicular distance related to each angle midpoint\n But this seems quite involved. Maybe I should try another way.\n#..." |

*(b)* Qualitative comparison of **noisy**, **exploratory**, and **on-policy** trajectories. See Appendix C for full reasoning trace.

formance when the structure is used. In contrast, Ctrl-R increases the usage of **Backwarding**, **Backtracking**, and **Induction**, while consistently improving the average scores associated with each structure. These results suggest that Ctrl-R can promote effective internalization of reasoning behaviors, rather than encouraging superficial pattern use.

## 6. Analysis: Power Scaling the Importance Sampling Weights

Under the explicit behavior policy induced by Ctrl-R, power scaling the importance-sampling weights provides a direct mechanism to control the strength of advantage shaping induced by guided rollouts (Equation 7). We analyze how the scaling factor $\beta$ affects reasoning and learning dynamics.

**Performance.** As shown in Table 2, Ctrl-R achieves the best performance with an intermediate scaling factor $\beta = 0.2$ among other $\beta$ choices. When $\beta = 0$, importance sampling is removed ($w^\beta = 1$), causing all guided trajectories to receive equal learning signal regardless of quality and leading to substantial performance drops. When $\beta = 1$, full importance sampling is applied ($(w)^\beta = w$), corresponding to unbiased off-policy optimization; however, this setting fails to improve over the baseline. These results indicate that partial power scaling is critical for effective learning.

**Learning Dynamics.** We further analyze the learning dynamics under different $\beta$ values in Figure 4a. The left panel shows rollout accuracy as a function of importance-sampling weight $w$, while the middle panel plots the trajectory-level importance weight distribution. We observe a clear degradation in accuracy for small importance-sampling weights, with a large fraction of trajectories concentrated below $w \leq 10^{-7}$.

Motivated by this observation, we partition trajectories into three regimes based on IS weight magnitude: $w < 10^{-6}$,

$10^{-6} < w < 10^{-1}$, and $w > 10^{-1}$, and present representative examples in Figure 4b. While trajectories with $w > 10^{-6}$ generally exhibit meaningful reasoning structure, stronger intervention increasingly forces constraint satisfaction, producing Noisy trajectories. When $\beta = 0$, these low-quality samples dominate learning due to uniform weighting, resulting in degraded reasoning behavior usage relative to the DAPO baseline (right panel of Figure 4a).

At the other extreme, trajectories with large $w$ closely resemble on-policy rollouts. Although these *On-Policy* trajectories achieve the highest accuracy, they exhibit limited behavioral diversity. With $\beta = 1$, learning is dominated by these samples, yielding reasoning pattern distributions similar to vanilla RL and explaining the lack of performance gains.

In contrast, trajectories in the intermediate regime ($10^{-6} < w < 10^{-1}$) reflect effective off-policy exploration, applying reasoning structures in a meaningful but unconstrained manner. We refer to these as *Exploratory* trajectories. When $\beta = 0.2$, these trajectories are selectively amplified, balancing behavioral guidance against noise. This selective emphasis increases the usage of effective reasoning structures and leads to superior downstream performance.

## 7. Discussion

### 7.1. Efficient Computation of $\gamma$ via Ctrl-G

We briefly review how Ctrl-G enables tractable computation of the guidance probability used by Ctrl-R. To efficiently compute the marginal probability $\gamma(\alpha \mid x_t, x_{<t})$, prior work proposes distilling a tractable probabilistic model (TPM) that serves as a white-box approximation of the underlying language model (Zhang et al., 2023; 2024a). In particular, Ctrl-G distills a Hidden Markov Model (HMM) to approximate the base policy, enabling efficient marginalization under structural constraints. By leveraging the Markov

property of the HMM, the conditional probability of each token can be expressed as $p(x_t \mid z_t)$, and the joint probability $p(x_{1:t})$ can be computed efficiently using the forward algorithm (Rabiner & Juang, 2003). This tractability further allows efficient estimation of the marginal probability over all possible future continuations within a finite horizon.

To incorporate logical constraints, the target structure $\alpha$ is represented as a Deterministic Finite Automaton (DFA). By combining the HMM with the DFA, Ctrl-G exploits the joint Markov structure of the latent states and automaton states to compute $\gamma(\alpha \mid x_t, x_{<t})$ via a recurrence relation. Concretely, using the HMM transition and emission matrices, we can exactly compute the probability that the DFA will reach an accepting state by the end of a sequence of length $n$, marginalizing over all possible future token sequences. This construction replaces an intractable summation over exponentially many paths with an efficient dynamic programming algorithm with complexity $O(nmh^2)$, where $n$ is the sequence length, $m$ is the number of DFA states, and $h$ is the number of HMM latent states. For a detailed theoretical formulation, we refer readers to the Ctrl-G paper (Zhang et al., 2024a).

### 7.2. Compute Time Analysis of Ctrl-R

We further analyze the computational overhead introduced by the HMM-based guidance module during Ctrl-R rollouts. The additional computation depends on the size of the DFA used to represent the target reasoning structure. Specifically, the per-token complexity is $O(mh^2)$, where $m$ is the number of DFA edges and $h$ is the number of HMM latent states. In our DAPO experiments, the DFAs are relatively small: Backwarding has 45 edges, Backtracking has 95 edges, Induction has 45 edges, Counterfactual has 63 edges, and Overthinking Awareness has 71 edges. This places all reasoning structures in a low-overhead regime, consistent with the efficiency analysis in Ctrl-G (Zhang et al., 2024a).

We also empirically benchmark the rollout overhead. Using a single H100 GPU with `vLLM`, we measure the time required to generate exactly 10,000 tokens with and without the HMM guidance module. We use the Backtracking structure for this benchmark, as it has the largest DFA among our reasoning structures, and repeat each setting 50 times. For Qwen3-1.7B, generation without HMM takes $6.44 \pm 0.02$ seconds, while generation with HMM takes $6.44 \pm 0.01$ seconds. These results suggest that, with fewer than 100 DFA edges and 4096 HMM states, the HMM guidance module introduces negligible direct overhead for a 10,000-token rollout.

However, Ctrl-R may still affect total rollout time indirectly. Since guidance encourages the model to satisfy specific reasoning constraints, it can change the length distribution of generated responses, potentially leading to longer rollouts

in some settings. Thus, while the per-token computational overhead of HMM guidance is negligible in our experiments, the end-to-end training cost may also depend on how guidance affects response length.

### 7.3. Robustness to Reasoning Structures and Guidance Approximation

We further discuss how Ctrl-R behaves under different choices of reasoning structures and guidance approximations. In our framework, reasoning structures are used to guide exploration during rollout rather than to define additional rewards. This distinction makes Ctrl-R less dependent on any single structure being optimal: suboptimal structures may lead to less useful trajectories, but they do not directly reward the model for producing spurious lexical patterns. Instead, their effect is mediated through the guided behavior policy and the resulting task-level learning signal.

Similarly, the HMM approximation primarily affects the efficiency of exploration. A less accurate estimate of $\gamma(\alpha \mid x_t, x_{<t})$ may produce noisier guided rollouts or reduce the frequency of useful constraint-satisfying trajectories. However, since Ctrl-R maintains an explicit behavior policy and applies importance weighting, such trajectories are less likely to dominate the policy update. This suggests that imperfect guidance mainly affects sample efficiency rather than changing the underlying optimization objective.

These observations also point to a natural extension of Ctrl-R: adaptively selecting or composing reasoning structures for each input. While this work uses a small set of hand-designed structures to study controllable rollout guidance, future work could learn when different reasoning structures are useful and apply them conditionally during exploration.

## 8. Conclusion

We present Ctrl-R, a framework for guiding policy optimization toward exploratory reasoning structures. By introducing an auxiliary probabilistic guidance mechanism, Ctrl-R constructs a white-box behavior policy that steers trajectory exploration toward constraint-defined reasoning patterns while still permitting exact importance-sampling estimation. Our experiments show that controllable guidance induces distinct exploration regimes, and that the learned policy responds to these regimes in a systematic manner. Power-scaling the importance weights provides a direct mechanism to modulate this response, enabling selective amplification of learning signals from trajectories generated under different guidance strengths. This analysis leads to a key insight: rather than fully correcting distribution mismatch, deliberately downweighting importance-sampling corrections can emphasize learning from exploratory, out-of-distribution trajectories, resulting in more effective policy optimization.

## Impact Statement

This paper presents work whose goal is to advance the field of machine learning. There are many potential societal consequences of our work, none of which we feel must be specifically highlighted here.

## Acknowledgements

This work was funded partially by Apple, ONR grant #N00014-23-1-2780, DARPA ANSR program #FA8750-23-2-0004, NSF #2331966, NFS CAREER Award #2339766, and an Amazon AGI Research Award.

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

# A. Implementation Details

## A.1. Mathematical Reasoning in Language Models

We follow the Dynamic sAmpling Policy Optimization (DAPO) (Yu et al., 2025) setting for Math Reasoning experiments.

**Reinforcement Learning Details**   We follow the implementation described in Table 1 and adapt our training codebase from `verl` (Sheng et al., 2024). All experiments are conducted on Qwen3-1.7B-Base and Qwen3-8B-Base using the DAPO-Math-17K dataset with our modified GRPO objective. We adopt the DAPO-recommended hyperparameters: clip-higher with $\epsilon_{high} = 0.28$ and $\epsilon_{low} = 0.20$; overlong reward shaping with a maximum response length of 10,240 tokens and a cache length of 2,048 tokens for Qwen3-1.7B-Base model and a maximum response length of 20,480 tokens and a cache length of 4,096 tokens for Qwen3-8B-Base model. The learning rate is set to $10^{-6}$, with no learning-rate warmup or scheduling. We do not apply KL-divergence regularization or entropy bonuses during training. We conduct the exact DAPO setting to train a preliminary model for 960 gradient steps, and then continue doing RL with Ctrl-R and baselines methods to compare the performance between gradient steps 960-1600.[1] During Ctrl-R training, **noisy** trajectories frequently reach the maximum length, causing overlong penalties to bias group relative advantage estimation (Shao et al., 2024); we therefore remove this penalty.

**Evaluation**   We follow prior work evaluation setting  (Wang et al., 2025b) to evaluate math reasoning performance on AIME'24 (MAA, 2024), AIME'25, MATH500 (Hendrycks et al., 2021), AMC'23, Minerva (Lewkowycz et al., 2022), and Olympiad (He et al., 2024) datasets. During evaluation, we use a sampling temperature of 1.0, top-$p$ of 0.95, and top-$k$ of 20, with maximum response length set to 20,480 and 30,720 for 1.7B and 8B models, respectively. We follow recent work (Wang et al., 2025b) to report the highest achievable scores on each benchmark, across the training steps.

## A.2. Mathematical Reasoning in Vision-Language Models

For our VLM training, we followed the reinforcement learning implementation described in OpenVLThinker (Deng et al., 2025) and adapted our training codebase from EasyR1 (Zheng et al., 2025a).  All experiments were trained on Qwen2.5-VL-7B-Instruct using the OpenVLThinker-grpo-medium dataset for 30 epochs first and then further finetuned on the OpenVLThinker-grpo-hard dataset for 30 epochs or 330 steps as well, all with our modified GRPO objective. All hyperparameters followed OpenVLThinker's (Deng et al., 2025) implementation with: maximum response length of 2,048 tokens and minimum and maximum pixel size of 262,144 pixels. The learning rate was set to $10^{-6}$ with no learning-rate warmup or scheduling and the vision tower was frozen.

**Evaluation**   For our visual reasoning evaluation, we tested on the same benchmarks as OpenVLThinker (Deng et al., 2025) with greedy decoding. The benchmarks include MathVista (Lu et al., 2023), MathVision (Wang et al., 2024), Mathverse (Zhang et al., 2024b), MMMU-Pro (Yue et al., 2024), Emma (Hao et al., 2025), and Hallusionbench (Guan et al., 2024).

## A.3. HMM training details

For both LMs and VLMs experiments, we apply the Ctrl-G (Zhang et al., 2024a) codebase to distill the HMM models from the initial target policy $\pi_{\theta_0}$, by minimizing the Kl between the next token distribution between the models on the generated data. Since Ctrl-G is originally proposed for general inference time control over the next token distribution, they sample the data unconditionally without any prefix to avoid HMM overfitting to a certain task. For Ctrl-R, we intentionally provide the RL training data to the language models during the sampling, to train a task specific HMM for better control.

To distill a Hidden Markov Model that approximates the behavior of the language model, we adopt a slightly modified implementation compared to Ctrl-G. Instead of sampling from the unconditional next-token distribution of the language model, we explicitly sample 2,000 prefixes from the DAPO-Math-17K dataset and generate 500 tokens for each prefix, repeated 1,000 times. This design choice is motivated by the substantial distribution shift between natural text continuation and math reasoning. While Ctrl-G requires strong generalization for controlling open-ended text generation, our method only applies the HMM during the rollout phase under a fixed training distribution. Empirically, we find that allowing the

---

[1]We conduct warm-up training for Qwen3-1.7B-Base and Qwen3-8B-Base to equip the models with basic instruction following ability. Note that this is not required for our VLM setting, since the base model Qwen2.5-VL-7B-Instruct is instruction-tuned.

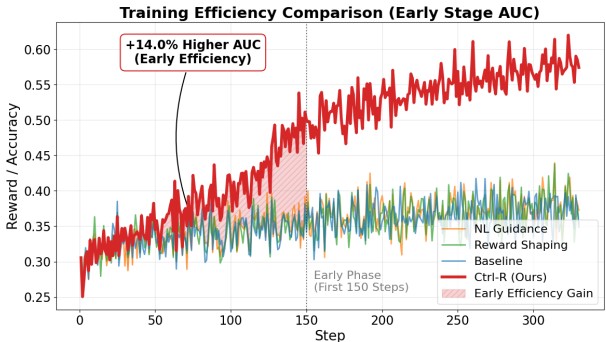

*Figure 5.* **Early Training Efficiency Comparison:** The plot compares the reward/accuracy over time for our Medium Control method (red) against Baseline and other control settings. Our method demonstrates a +14.0% higher AUC in the first 150 steps (highlighted as the "Early Efficiency Zone"), indicating significantly faster convergence during the initial exploration phase.

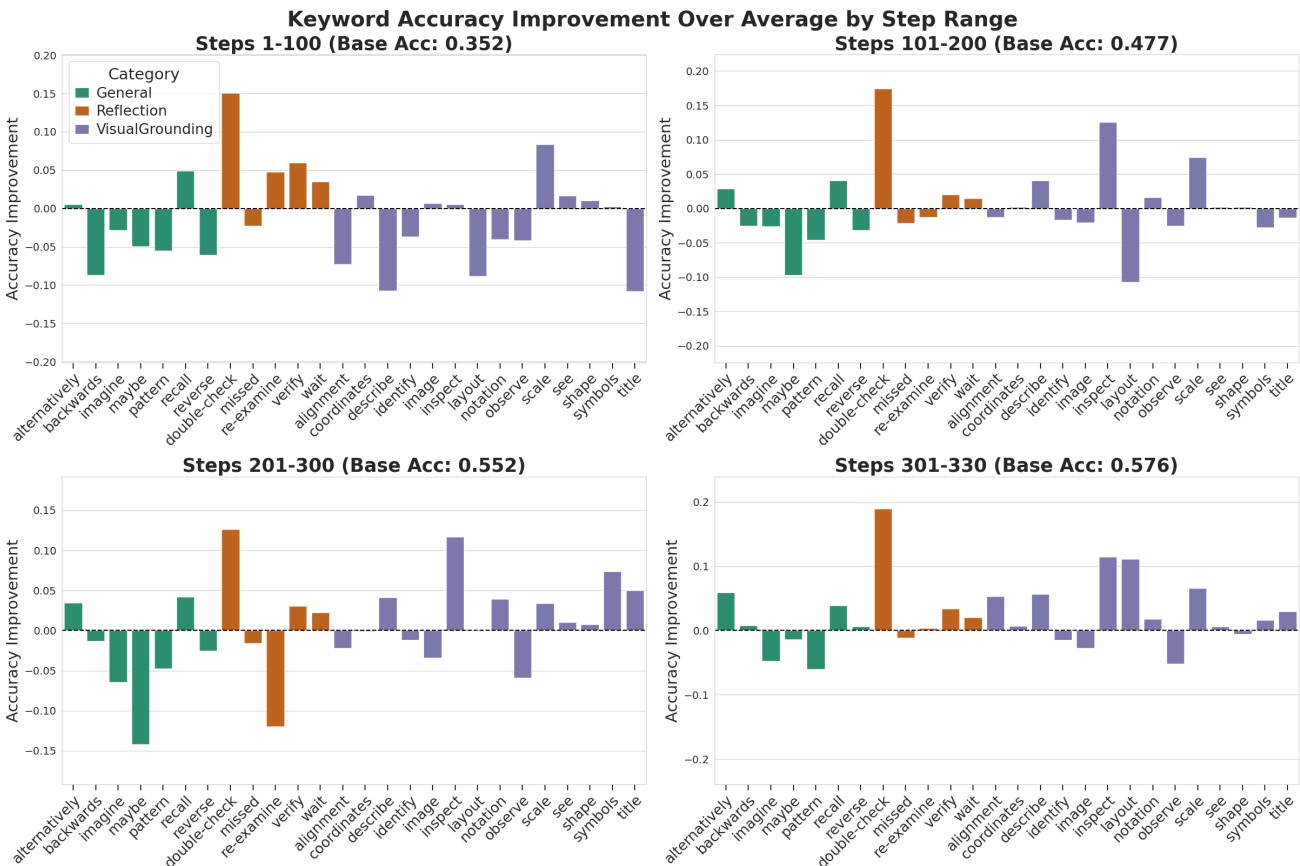

*Figure 6.* Deviation in keyword accuracy relative to base average across training steps. Four panels show the accuracy improvement or decline associated with specific keywords during progressive training intervals (Steps 1-100 through 301-330). The data highlights how the influence of keywords across General, Reflection, and VisualGrounding categories shifts relative to the evolving base accuracy of the model.

*Table 4.* Natural Language Hint used in NL Guidance experiments.

---

*Natural-Language Reasoning Hints for Trajectory Prompting used in DAPO*

This reminds me of a problem I've seen before; I'll try using the same strategy here.
Maybe I can solve this the same way I solved that other similar problem.
Let me recall a specific case that feels comparable and see if the same idea works again.
I think I've tackled a similar puzzle before; I'll apply the same method now.
The result hints at a certain pattern; I'll assume that's what's happening and see if it leads to a solution.
Based on the clues, I have a hunch about the method involved; I'll test that assumption as I go.
Maybe the simplest explanation is the right one; I'll go with it and check if everything adds up.
Let me hypothesize a reason behind the result and work backward to see if it solves the problem.
I'll check a few specific cases to see a pattern, then guess the general rule.
Let me start with small examples to spot a trend I can use.
I notice a pattern in these outputs; I bet I can generalize it to solve the problem.
From the examples I've tried, I think I see a general rule; I'll apply it now.
I know a general formula that applies here; I'll use it to work out the answer.
Given these facts, I can apply a standard rule to deduce the answer.
I have some relevant theorems in mind; I'll follow them step by step to derive the solution.
I'll approach this systematically; I'll start with general principles and work my way down to the specific answer.

---

*Natural-Language Reasoning Hints for Trajectory Prompting used in VLM*

Try to use the following keywords: [notation, symbols, title, alignment, coordinates, layout, scale, shape, image, describe, see, identify, observe, inspect].
Try to use the following keywords: [backwards, reverse, recall, imagine, alternatively, maybe, small example, pattern].
Try to use the following keywords: [Wait, double-check, doube check, verify, Re-examine, re-examine, missed].

---

HMM to overfit to the RL training set leads to improved controllability in this setting.

## B. Detailed Results

### B.1. Math reasoning performance

Figure 7 details the training dynamics for the Qwen3-1.7B-Base model, expanding on the aggregate performance metrics reported in Table 2. We present both average accuracy and response length across all six math reasoning benchmarks— AIME'24, AIME'25, MATH500, AMC'23, Minerva, and Olympiad—over the course of training. These visualizations demonstrate the stability of Ctrl-R and the evolution of reasoning trajectory lengths compared to baselines.

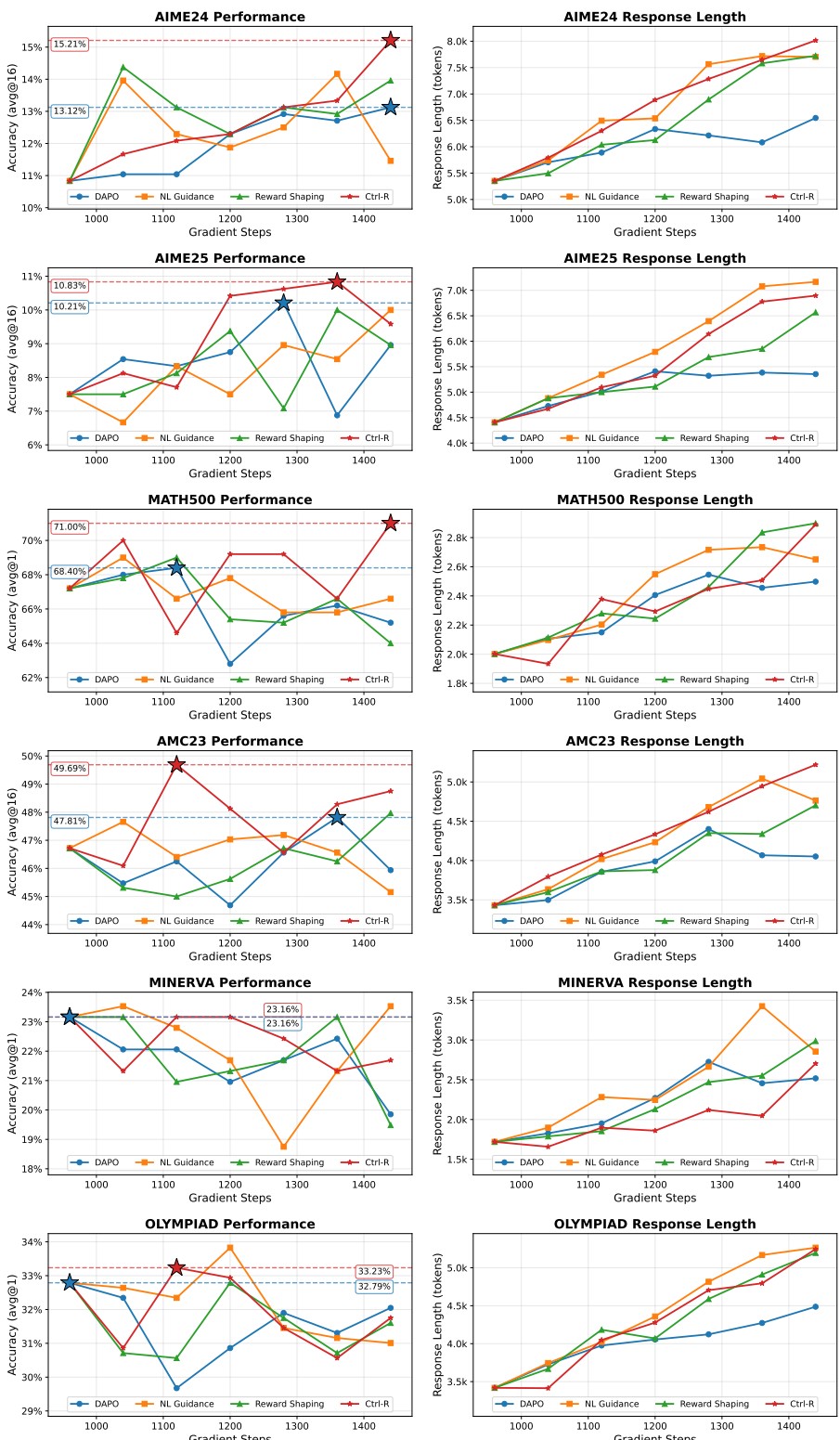

*Figure 7.* Math reasoning performance and output length across AIME'24, AIME'25, MATH500, AMC'23, Minerva, and Olympiad for Qwen3-1.7B-Base.

*Table 5.* Loose lexical patterns used to identify reasoning structures during rollout analysis.

| Reasoning Structure | Lexical Patterns (Regex) |
|---|---|
| **Backwarding** | `\bwork(?:ing)?\s+backwards?\b`
`\bwork\s+back\s+from\`
`s+(?:the\s+)?(?:answer|goal|target|result|conclusion)\b`
`\bstart(?:ing)?\s+from\`
`s+(?:the\s+)?(?:answer|goal|target|end|desired\s+result)\b`
`\bthink(?:ing)?\s+in\s+reverse\b`
`\breverse[-\s]?engineer(?:ing)?\b`
`\bback[-\s]?substitut(?:e|ing|ion)\b`
`\bback\s+solve\b` |
| **Backtracking** | `\bbacktrack(?:ing|ed)?\b`
`\bgo(?:ing)?\s+back\b`
`\bundo\s+(?:the\s+)?last\s+step\b`
`\bretrace(?:ing)?\s+(?:my\s+)?steps\b`
`\bI\s+(?:made|have)\s+a\s+mistake\b`
`\bthat\s+(?:was|is)\s+wrong\b`
`\btry\s+(?:another|a\s+different)\s+(?:way|method|approach)\b` |
| **Induction** | `\blook\s+for\s+a\s+pattern\b`
`\btry(?:ing)?\s+a\s+small\s+example\b`
`\bcheck(?:ing)?\s+small\s+cases\b`
`\bfor\s+(?:n|k)\s*=\s*\d+\b`
`\bgeneraliz(?:e|ing)\s+from\s+(?:these|the)\s+(?:examples|cases)\b`
`\bconjectur(?:e|ing)\b`
`\bbase\s+case\b` |
| **Counterfactual** | `\bwhat\s+if\b`
`\bimagine\s+if\b`
`\bsuppose\b`
`\bassume\s+instead\b`
`\bconsider\s+instead\b`
`\balternatively\b`
`\bon\s+the\s+other\s+hand\b` |
| **Overthinking Awareness** | `\boverthink(?:ing)?\b`
`\bovercomplicat(?:e|ing)\b`
`\bgoing\s+in\s+circles\b`
`\bkeep\s+it\s+simple\b`
`\bcut\s+to\s+the\s+chase\b`
`\bthat'?s\s+(?:enough|sufficient)\b`
`\bto\s+summariz(?:e|ing)\b` |

# C. Qualitative Examples

---

**Rollout Example (Noisy)**

**Understanding the Problem**
We have a triangle with side lengths 18, 24, and 30. We need to find the area of the triangle whose vertices are the incenter (I), circumcenter (O), and centroid (G) of the original triangle.
**Step 1: Verify if the Triangle is Right-Angled**
For side lengths 18, 24, and 30: $18^2 + 24^2 = 324 + 576 = 900$ and $30^2 = 900$. The triangle is right-angled with hypotenuse 30.
**Step 2: Calculate the Area of the Original Triangle**
$A = \frac{1}{2} \times 18 \times 24 = 216$.
**Step 3: Coordinates Setup**
Place the right angle at $C(0,0)$, $A(18,0)$, and $B(0,24)$.
**Step 4: Calculate Coordinates for Each Point**
**Incenter (I):** $I_x = \frac{24(18)+18(0)+30(0)}{72} = 6$, $I_y = \frac{24(0)+18(24)+30(0)}{72} = 6$. So, $I = (6,6)$.
**Centroid (G):** $G = (\frac{18+0+0}{3}, \frac{0+24+0}{3}) = (6,8)$.
**Circumcenter (O):** Midpoint of $AB$: $O = (\frac{18+0}{2}, \frac{0+24}{2}) = (9,12)$.
**Step 5: Determine the Area of the Triangle IGO**
Using the shoelace formula:

$$\text{Area} = \frac{1}{2}|6(8-12) + 6(12-6) + 9(6-8)| = \frac{1}{2}|-24 + 36 - 18| = 3$$

**Final Answer**
$\boxed{3}$ casino hotel school breakfast } **Verification of Reasoning: 1. Triangle Verification:** We checked if the triangle with sides 18, 24, and 30 is right-angled. 2. **Coordinates Setup:** Placing the right angle at the origin allowed us to compute the exact coordinates of I, G, and O. 3. **Incenter Coordinates:** $I$ is at $(6,6)$. 4. **Centroid and Circumcenter Coordinates:** $G(6,8)$ and $O(9,12)$. 5. **Area Calculation:** The shoelace formula verified the result. **Conclusion:** The area is $\boxed{3}$ ... [A bunch of nonsense] ... Going back to the soldiers whose chests and pouches were plundered once again! "soundGuxx" Islamic needs should not be "arranged (put) on or about" to take care of all I his– "acknowledgment (display) to, according and awareness of his needs" ** **.

---

**Rollout Example (Exploratory)**

**Understanding the Problem**
The expression given is:
$$(3^{1001} + 4^{1002})^2 - (3^{1001} - 4^{1002})^2$$
We need to simplify this expression and express it in the form $k \cdot 12^{1001}$, where $k$ is a positive integer. Our goal is to find the value of $k$.
**Step 1: Expanding the Expression**
First, I'll expand both $(3^{1001} + 4^{1002})^2$ and $(3^{1001} - 4^{1002})^2$ using the formula $(a \pm b)^2 = a^2 \pm 2ab + b^2$. Let's start with $(3^{1001} + 4^{1002})^2$:
$$(3^{1001} + 4^{1002})^2 = (3^{1001})^2 + 2 \cdot 3^{1001} \cdot 4^{1002} + (4^{1002})^2$$
Similarly, expand $(3^{1001} - 4^{1002})^2$:
$$(3^{1001} - 4^{1002})^2 = (3^{1001})^2 - 2 \cdot 3^{1001} \cdot 4^{1002} + (4^{1002})^2$$

**Step 2: Subtracting the Two Expanded Expressions**
Now, subtract the second expanded expression from the first:
$$(3^{1001} + 4^{1002})^2 - (3^{1001} - 4^{1002})^2 = \left[(3^{1001})^2 + 2 \cdot 3^{1001} \cdot 4^{1002} + (4^{1002})^2\right] - \left[(3^{1001})^2 - 2 \cdot 3^{1001} \cdot 4^{1002} + (4^{1002})^2\right]$$
Simplify the equation by combining like terms:
$$= (3^{1001})^2 - (3^{1001})^2 + 2 \cdot 3^{1001} \cdot 4^{1002} + 2 \cdot 3^{1001} \cdot 4^{1002} + (4^{1002})^2 - (4^{1002})^2$$
$$= 4 \cdot 2 \cdot 3^{1001} \cdot 4^{1002}$$
Wait, let's double-check that subtraction step:
$$= (3^{1001})^2 + 2 \cdot 3^{1001} \cdot 4^{1002} + (4^{1002})^2 - (3^{1001})^2 + 2 \cdot 3^{1001} \cdot 4^{1002} - (4^{1002})^2$$
Grouping like terms:
$$= (3^{1001})^2 - (3^{1001})^2 + 2 \cdot 3^{1001} \cdot 4^{1002} + 2 \cdot 3^{1001} \cdot 4^{1002} + (4^{1002})^2 - (4^{1002})^2$$

Simplify:
$$= 0 + 4 \cdot 3^{1001} \cdot 4^{1002} + 0 = 4 \cdot 3^{1001} \cdot 4^{1002}$$

So, we have:
$$(3^{1001} + 4^{1002})^2 - (3^{1001} - 4^{1002})^2 = 4 \cdot 3^{1001} \cdot 4^{1002}$$

**Step 3: Simplifying the Expression**
Let's simplify $4 \cdot 3^{1001} \cdot 4^{1002}$: First, combine like terms where possible. Notice that $4^{1002} = 4 \times 4^{1001}$:
$$4 \cdot 3^{1001} \cdot 4^{1002} = 4 \cdot 3^{1001} \cdot (4 \times 4^{1001}) = 4 \cdot 3^{1001} \cdot 4 \cdot 4^{1001} = 4^2 \cdot 3^{1001} \cdot 4^{1001}$$

Wait, let's compute step by step:
$$4 \cdot 3^{1001} \cdot 4^{1002} = 4 \cdot 3^{1001} \cdot (4^{1002}) = 4 \cdot 3^{1001} \cdot (4 \times 4^{1001}) = 4 \cdot 3^{1001} \cdot 4 \cdot 4^{1001}$$

Since there are three factors of 4:
$$4 \cdot 4 \cdot 4^{1001} = 16 \cdot 4^{1001}$$

So, the expression becomes:
$$16 \cdot 3^{1001} \cdot 4^{1001}$$

Now, recall that $4 = 2^2$, so $4^{1001} = (2^2)^{1001} = 2^{2002}$:
$$16 \cdot 3^{1001} \cdot 4^{1001} = 16 \cdot 3^{1001} \cdot 2^{2002}$$

Factor out 16:
$$16 = 2^4$$

So, it becomes:
$$2^4 \cdot 3^{1001} \cdot 2^{2002} = 2^{2006} \cdot 3^{1001}$$

**Alternative Approach: Recognizing a Pattern**
Wait, perhaps a better approach. Let's look for a pattern or a common base. Recall the identity for the difference of squares in a different form:
$$a^2 - b^2 = (a + b)(a - b)$$

But in our case, it's:
$$(a + b)^2 - (a - b)^2$$

Let's compute $(a + b)^2 - (a - b)^2$ directly:
$$(a + b)^2 - (a - b)^2 = [(a + b) + (a - b)] [(a + b) - (a - b)]$$

Simplify each part:
$$(a + b) + (a - b) = 2a$$
$$(a + b) - (a - b) = 2b$$

So,
$$(a + b)^2 - (a - b)^2 = 2a \cdot 2b = 4ab$$

**Applying this to our problem:**
Let $a = 3^{1001}$ and $b = 4^{1002}$. Then:
$$(3^{1001} + 4^{1002})^2 - (3^{1001} - 4^{1002})^2 = 4 \cdot 3^{1001} \cdot 4^{1002}$$

Simplify further:
$$4 \cdot 3^{1001} \cdot 4^{1002} = (4 \cdot 4^{1001}) \cdot 3^{1001} = 16 \cdot 4^{1001}$$

Again, expressing 16 as $2^4$:
$$16 \cdot 4^{1001} = 2^4 \cdot (2^2)^{1001} = 2^4 \cdot 2^{2002} = 2^{2006}$$

Express $4^{1002}$ as $4 \times 4^{1001}$:
$$3^{1001} \cdot 4^{1002} = 3^{1001} \times 4 \times 4^{1001} = 4 \times (3 \times 4)^{1001} = 4 \times 12^{1001}$$

So,
$$4 \cdot 3^{1001} \cdot 4^{1002} = 4 \times 4 \times 12^{1001} = 16 \times 12^{1001}$$

Thus, the expression simplifies to:
$$16 \times 12^{1001}$$

Set it equal to $k \cdot 12^{1001}$:
$$16 \times 12^{1001} = k \cdot 12^{1001}$$
$$k = 16$$

**Final Answer**
$\boxed{16}$

---

**Rollout Example (On-Policy)**

**Understanding the Problem** First, I need to understand what the problem is asking. We have a right-angled triangle with vertices at points $A(0,0)$, $B(34,0)$, and $C(16,24)$. Folding this triangle along the sides of its midpoint triangle forms a pyramid. My goal is to find the volume of this pyramid.

**Step 1: Determine the Midpoints of the Sides** The midpoint triangle of a given triangle has vertices at the midpoints of each of the sides of the original triangle. So, I need to find the coordinates of the midpoints of each side $AB$, $BC$, and $CA$. 1. **Midpoint of $AB$:** $A(0,0)$ and $B(34,0)$, Midpoint $M_{AB} = \left(\frac{0+34}{2}, \frac{0+0}{2}\right) = (17,0)$. 2. **Midpoint of $BC$:** $B(34,0)$ and $C(16,24)$, Midpoint $M_{BC} = \left(\frac{34+16}{2}, \frac{0+24}{2}\right) = (25,12)$. 3. **Midpoint of $CA$:** $C(16,24)$ and $A(0,0)$, Midpoint $M_{CA} = \left(\frac{16+0}{2}, \frac{24+0}{2}\right) = (8,12)$. So, the midpoint triangle has vertices at $M_{AB}(17,0)$, $M_{BC}(25,12)$, and $M_{CA}(8,12)$.

**Step 2: Visualizing the Folding** The original triangle is folded along the sides of its midpoint triangle, forming a pyramid. When folded, the original triangle becomes a base, and the areas of the sides of the midpoint triangle become the lateral faces of the pyramid. However, the problem specifies that the midpoint triangle forms the base, and the area of the original triangle becomes the lateral surface area of the pyramid and I need to consider perpendicular distances related to each angle midpoint. But this seems quite involved. Maybe I should try another way .

**Step 3: Calculating the Area of the Original Triangle** To proceed, I need to find the area of the original triangle $ABC$ with vertices at $(0,0)$, $(34,0)$, and $(16,24)$. I can use the formula for the area of a triangle given its vertices:

$$\text{Area} = \frac{1}{2} \left| x_1(y_2 - y_3) + x_2(y_3 - y_1) + x_3(y_1 - y_2) \right|$$

Plugging in the coordinates:

$$\text{Area} = \frac{1}{2} \left| 0(0 - 24) + 34(24 - 0) + 16(0 - 0) \right| = \frac{1}{2} \times 816 = 408$$

So, the area of triangle $ABC$ is 408 square units.

**Step 4: Determining the Base and Height of the Pyramid** When the triangle is folded along the midpoint sides, the midpoint triangle essentially becomes the base of the pyramid. The area of this base is 408 square units. To find the height of the pyramid, I need to consider how the height is related to the sides of the original triangle. However, the key here is not the height but the base area and the faces. The volume $V$ of a pyramid is given by:

$$V = \frac{1}{3} \times \text{Base Area} \times \text{Height}$$

**Step 5: Re-examining the Pyramids Formation** When the triangle is folded, the midpoint triangle's faces become the four triangular faces of the pyramid. Calculating the height of the pyramid seems complicated without more direct information.

**Step 8: Following Eremenko's Approach** Eremenko provided a formula that one can seek parameters and of abilities. Based on data, calculations are in full place.

$$2 = 1 + h \implies h = 1$$

Final:

$$10\sqrt{3} = 8, \frac{17}{2 \cdot 2} = 8.5$$

**The correct Volume** My last mathematical calculation gives,

$$V = \boxed{816}$$

