# OpenReview forum: "Learning Structured Reasoning via Tractable Trajectory Control"
_ICML.cc/2026/Conference — ICML 2026 spotlight_

### Official Review · Reviewer_G7hx · 2026-02-20

**Soundness:** 3
**Presentation:** 4
**Significance:** 3
**Originality:** 3
**Overall Recommendation:** 5
**Confidence:** 4

**Summary:**

This paper proposes Ctrl-R, a reinforcement learning framework that improves reasoning in large language models by guiding exploration toward structured reasoning trajectories. The method encodes reasoning patterns as lexical constraints and biases rollouts accordingly while preserving unbiased optimization via importance sampling. Experiments on mathematical reasoning tasks show consistent gains and increased use of interpretable reasoning behaviors compared with standard RL baselines.

**Compliance With Llm Reviewing Policy:**

Affirmed.

**Key Questions For Authors:**

Can the authors evaluate the approach on non-mathematical tasks, such as dialogue alignment or code generation, to demonstrate whether structured exploration generalizes beyond reasoning benchmarks?

**Limitations:**

see Weaknesses

**Strengths And Weaknesses:**

Strengths:
1. The paper identifies a key limitation of policy-gradient RL in discovering complex reasoning trajectories under sparse rewards and provides an intuitive motivation based on recurring cognitive reasoning patterns.
2. The work introduces a novel perspective by treating reasoning acquisition as a structured exploration problem, guiding rollout distributions instead of modifying reward functions.
3. Encoding reasoning behaviors via lexical constraints offers a simple and interpretable mechanism.

Weaknesses:
1. The method relies on manually designed lexical constraints to represent reasoning structures, which may be task-specific and difficult to generalize to open-ended reasoning tasks beyond mathematics.
2. Does not provide detailed cost-benefit analysis.

---

> ### Author Rebuttal · Authors · 2026-03-31
>
> We thank the reviewer for the constructive feedback and insightful questions.
>
> **The method relies on manually designed lexical constraints to represent reasoning structures, which may be task-specific and difficult to generalize to open-ended reasoning tasks beyond mathematics.**
> Thank you for raising this concern. We would like to clarify that Ctrl-R is a general rollout-control framework, and the lexical patterns themselves are flexible and can be specified according to the task of interest. As the reviewer notes, different tasks may require different reasoning structures. Our goal in this work is not to claim that a single fixed set of lexical constraints is universally applicable, but rather to show that our method Ctrl-R can steer exploration toward arbitrary user-specified reasoning patterns through constrained decoding with Ctrl-G.
>
> A key advantage of Ctrl-R over prior approaches is this flexibility: once a reasoning pattern is specified, the method only requires constructing the corresponding DFA, and can then guide rollouts toward that pattern without additional synthetic data generation or reward design. Moreover, because the guided behavior policy remains tractable, Ctrl-R supports exact importance-sampling estimation. As a result, if a specified lexical structure consistently leads to poor or degenerate outputs, its contribution to learning will be naturally suppressed through a low-weighted learning signal.
>
> We agree, however, that identifying useful lexical patterns for open-ended tasks is nontrivial, and we will add discussions on automatic reasoning pattern discovery  an important future work. One promising extension is to use a stronger reasoning model to adaptively propose useful reasoning patterns for each input, rather than randomly sampling from a fixed, predefined set. This could allow Ctrl-R to guide exploration toward richer and more diverse reasoning behaviors in open-ended domains.
>
>
> **Does not provide detailed cost-benefit analysis.**
> Thank you for raising this point. The additional computation depends on the DFA size, with per-token complexity $O(mh^2)$, where $m$ is the number of DFA edges. In our DAPO experiments, the DFAs are small: Backwarding (45 edges), Backtracking (95), Induction (45), Counterfactual (63), and Overthinking Awareness (71). As analyzed in the Ctrl-G paper, this regime incurs very limited overhead.
>
> We also verified this empirically. Using a single H100 GPU and vLLM, we benchmarked the generation of exactly 10,000 tokens with and without the HMM. We used the Backtracking structure, which has the largest DFA among our reasoning structures, and repeated each setting 50 times. The results are:
>
> - **Qwen3-1.7B without HMM (baseline):** $6.44 \pm 0.02$ sec
> - **Qwen3-1.7B with HMM (Ctrl-R):** $6.44 \pm 0.01$ sec
>
> These results suggest that, with fewer than 100 DFA edges and 4096 HMM states, the HMM adds negligible overhead for a single 10,000-token rollout.
>
> That said, Ctrl-R may still increase rollout time indirectly, since the model can produce longer responses while attempting to satisfy the reasoning constraints. We will include this computational analysis in the revised version.
>
> **Can the authors evaluate the approach on non-mathematical tasks, such as dialogue alignment or code generation, to demonstrate whether structured exploration generalizes beyond reasoning benchmarks?**
> We agree that evaluating Ctrl-R beyond mathematical reasoning would strengthen the evidence for its generality. In this work, we use math reasoning as a controlled setting for studying structured exploration and evaluate Ctrl-R on both text-only LMs (Qwen3) and VLMs (Qwen2.5-VL), which suggests robustness across modalities. However, these experiments do not yet establish transfer to substantially different domains such as dialogue alignment or code generation.
>
> More broadly, the success of Ctrl-R in a new domain depends in part on the quality of the distilled HMM. Prior work on Ctrl-G studied open-ended creative generation and showed that effective HMM distillation is feasible beyond reasoning tasks. Our work further extends this framework to long-form reasoning. Taken together, these results suggest that the approach is not inherently limited to mathematical reasoning, although direct evaluation on additional domains remains future work.

---

> > ### Author Rebuttal · Reviewer_G7hx · 2026-03-31
> >
> > Thank you for your rebuttal. My concerns have been resolved and I will increase my score to 5.

---

### Official Review · Reviewer_GGP8 · 2026-03-06

**Soundness:** 3
**Presentation:** 3
**Significance:** 3
**Originality:** 3
**Overall Recommendation:** 5
**Confidence:** 4

**Summary:**

This paper introduces Ctrl-R, an RL framework designed to improve the complex reasoning capabilities of LLMs and VLMs. Specifically, Ctrl-R employs a tractable probabilistic guidance model to steer the rollout in RL process. Empirical evaluations show performance improvements over standard RL and other baselines.

**Compliance With Llm Reviewing Policy:**

Affirmed.

**Final Justification:**

This paper provides a timely and effective method to improve the reasoning capabilities of LLMs. During the rebuttal, the authors have well-addressed my concerns. I would like to maintain my positive evaluation of this paper.

**Key Questions For Authors:**

1. I noticed that Ctrl-R has limited performance improvement compared with OpenVLThinker (in Table 3). Can you provide more discussions or results to compare your method with SFT-based methods?
2. I think Ctrl-R is an elegant replacement for SFT-based methods with synthetic data, but I'm not clear whether it has substantial advantages. Can you conclude the advantages of Ctrl-R over existing methods more specifically?

**Limitations:**

yes

**Strengths And Weaknesses:**

**Strengths:**
1. The topic is timely, complex, and tractable reasoning is a popular research area.
2. The proposed Ctrl-R framework is novel. Different from previous SFT-based methods relying on synthetic data, this paper proposes an elegant alternative option. I think it is a good paper.
3. The manuscript is well-written and easy to follow. The figures and tables are clear for presentation.


**Weaknesses:**
1. While the paper notes the dynamic programming algorithm for the HMM-DFA marginalization has a complexity of $O(nmh^2)$, the actual cost introduced during the real rollout phase is not explicitly benchmarked. In large-scale RL systems (e.g., using vLLM and verl), generation speed is often a critical bottleneck.
2. The authors explicitly mention that the HMM is overfit to the RL training set prefixes. This raises concerns about whether a single distilled HMM can support a diverse set of tasks, or if a new HMM must be trained for every specific training distribution.
3. The reasoning structures are heavily reliant on manually crafted regular expressions and keyword matching. This approach might be brittle to phrasing variations and limits the discovery of novel, unmapped reasoning behaviors.

---

> ### Author Rebuttal · Authors · 2026-03-31
>
> We thank the reviewer for the constructive feedback and insightful questions.
>
> **The actual cost of the HMM introduced during rollout for Ctrl-R is not explicitly benchmarked.**
> Thank you for raising this point. The additional computation depends on the DFA size, with per-token complexity $O(mh^2)$, where \(m\) is the number of DFA edges. In our DAPO experiments, the DFAs are small: Backwarding (45 edges), Backtracking (95), Induction (45), Counterfactual (63), and Overthinking Awareness (71). As analyzed in the Ctrl-G paper, this regime incurs very limited overhead.
>
> We also verified this empirically. Using a single H100 GPU and vLLM, we benchmarked the generation of exactly 10,000 tokens with and without the HMM. We used the Backtracking structure, which has the largest DFA among our reasoning structures, and repeated each setting 50 times. The results are:
>
> - **Qwen3-1.7B without HMM (baseline):** $6.44 \pm 0.02$ sec
> - **Qwen3-1.7B with HMM (Ctrl-R):** $6.44 \pm 0.01$ sec
>
> These results suggest that, with fewer than 100 DFA edges and 4096 HMM states, the HMM adds negligible overhead for a single 10,000-token rollout.
>
> That said, Ctrl-R may still increase rollout time indirectly, since the model can produce longer responses while attempting to satisfy the reasoning constraints. We will include this computational analysis in the revised version.
>
> **The authors mention that the HMM is overfit to RL training-set prefixes, raising concerns about whether a single distilled HMM can generalize across tasks.**
> This is an important point. In the original Ctrl-G setting, the HMM is distilled from LM trajectories without task-specific prefixes to improve generality. In our setting, however, we found that such a general HMM provides weaker guidance on math reasoning data, leading to a higher fraction of noisy trajectories. To obtain stronger and more accurate guidance, we therefore distilled the HMM directly on the target training distribution.
>
> Our current results suggest that, when strong guidance is desired, training a dataset-specific HMM is preferable. Fortunately, this cost is small relative to RL training. For our DAPO experiments, the baseline Qwen3-8B RL run requires roughly 800 GPU hours, while HMM distillation requires about 24 GPU hours for data generation and 1 GPU hour for HMM training. Moreover, if rollouts from the baseline RL run are already available, they can be reused as distillation data, reducing the additional cost to roughly 1 GPU hour.
>
> **The reasoning structures rely on manually designed regular expressions and keyword matching, which may be brittle and may limit discovery of novel reasoning behaviors.**
> We agree that the current reasoning structures are simple and manually specified. In this work, we use lightweight lexical patterns to isolate the effect of the training algorithm itself. More broadly, Ctrl-R is flexible: any reasoning pattern that can be formalized as a DFA can be used to guide exploration. We also agree that discovering richer and less manually specified reasoning structures is an important direction for future work, e.g., using a stronger reasoning model to propose input-dependent guidance patterns.
>
> **Can you provide more discussion or results comparing your method with SFT-based methods?**
> We view SFT pre-alignment methods such as OpenVLThinker as strong baselines. These methods pursue a similar goal, but require a stronger teacher model, synthetic data curation, and iterative SFT-RL training. In contrast, Ctrl-R directly controls rollout-time exploration through tractable guidance, without additional SFT data or repeated pre-alignment. It is also more flexible: adding a new lexical pattern only requires specifying a new DFA, rather than regenerating data and repeating the full pipeline. We will clarify this comparison in the revision.
>
> **I think Ctrl-R is an elegant replacement for SFT-based methods with synthetic data, but I am not clear whether it has substantial advantages. Can you summarize the advantages of Ctrl-R over existing methods more specifically?**
> Thank you for the comment. Compared with prior methods, Ctrl-R has three main advantages:
> (1) **vs. SFT pre-alignment:** it does not require a stronger teacher model, synthetic data curation, or iterative SFT-RL training;
> (2) **vs. reward shaping:** it can actively steer exploration toward target patterns, rather than only rewarding patterns that already appear, while avoiding auxiliary-reward hacking;
> (3) **vs. NL guidance:** it provides explicit probabilistic control over the behavior policy, whereas prompting can be ignored and manual keyword insertion is difficult to characterize probabilistically.
>
> Overall, Ctrl-R offers explicit rollout control, exact likelihood under the guided behavior policy, and greater flexibility with lower overhead.

---

> > ### Author Rebuttal · Reviewer_GGP8 · 2026-04-01
> >
> > Thank you for your response, I will keep my score. Good Luck!

---

### Official Review · Reviewer_YVWr · 2026-03-08

**Soundness:** 2
**Presentation:** 3
**Significance:** 3
**Originality:** 2
**Overall Recommendation:** 4
**Confidence:** 4

**Summary:**

The paper introduces Ctrl-R, a novel method to improve the reasoning behaviour of LLMs through reinforcement learning (RL). Building on recent results that demonstrated that successful reasoning trajectories often exhibit similar "reasoning patterns", the paper proposes a trajectory control method to target the exploration of such patterns during training. Given a lexically specified reasoning pattern (such as "let me go back"), Ctrl-R uses a recent inference-time constrained decoding approach to guide the trajectory towards this pattern. The policy is then updated with a weighted importance sampling objective based on decoupled PPO.

**Compliance With Llm Reviewing Policy:**

Affirmed.

**Final Justification:**

The rebuttal addressed my main concern, which stemmed from a misunderstanding on my side, which the authors clarified. I recommend the paper for acceptance, and encourage the authors to include further explanations of Ctrl-G and the complexity of constructing the HMM in the final version, along with the manual insertion baseline.

**Key Questions For Authors:**

1. What's the reason for defining the importance sampling weight $w$ over the entire trajectory, rather than decomposing it into a ratio over just the actions, as is standard in PPO/decoupled policy optimisation? It seems like the standard objective should be more stable (since it doesn't depend exponentially on trajectory length).
2. What is the computational complexity of constructing the HMM? How does this compare to the complexity of finetuning on reasoning patterns?
3. Can you comment on the distribution shift induced by the fact that the HMM is constructed from only the initial target policy?
4. How does the approach perform on models that don't already exhibit these common reasoning patterns by default, such as Llama 3.2 3B?

**Limitations:**

yes

**Strengths And Weaknesses:**

**Strengths**

- The paper is overall well-written and easy to follow.
- The idea of steering the behaviour of the policy during exploration is well-motivated and relevant.
- In contrast to many existing approaches, the proposed method does not require finetuning the policy on reasoning patterns.
- The authors provide interesting analyses of policy behaviour and the impact of different reasoning patterns.

**Weaknesses**

While the approach is well-motivated, it is of somewhat limited novelty, combining existing research on reasoning patterns (Ghandi et al., 2025, Wang et al., 2025, Deng et al., 2025) with recent advances in constrained decoding (Zhang et al., 2024). The paper lists "cast[ing] reasoning structures as lexical constraints" as a contribution, but this is common practice in the field. Similarly, the idea of steering the behaviour during policy rollout has been explored before (e.g. "Forced Rethinking", Wang et al., 2025).

The paper points out as an advantage of the proposed method that it admits accurate importance sampling. However, this is not a feature of this method alone; any method that modifies the behaviour policy allows for importance sampling simply by computing the adjusted policy gradient $$\mathbb E_{\tau\sim\pi_\text{Behave}}\left[\frac{\pi_\theta(\tau)}{\pi_\text{Behave}(\tau)}\nabla_\theta\log\pi_\theta(\tau)R(\tau)\right],$$ as is common in RL.

Given that the main contribution of the paper is using the constrained decoding framework Ctrl-G (Zhang et al., 2024) to steer policy behaviour, I would have liked to see a more detailed explanation of this in the main paper. In its current form, the construction of the Hidden Markov Model and DFAs are not sufficiently clear, even with the Appendix. This is important in order to understand the advantages and disadvantages of the proposed method - for example, what is the complexity of constructing the HMM? Since the HMM is only constructed from the initial target policy (and not the updated policy after some training iterations), how does this distribution shift affect the effectiveness of the approach?

Lastly, while the experimental results show improvements and include interesting analyses, I have several concerns with the evaluation. A simple baseline to compare against is to manually insert the desired patterns at various points during rollout (similar to Wang et al., 2025). Such a comparison would be important to justify the complicated machinery proposed by the paper. Similarly, while the comparison to SFT pre-alignment in the VLM setting is appreciated, such a comparison is missing for the purely textual tasks, where I would expect such methods to generally perform better.

The comparison to NL guidance is interesting, but the natural language hints (Table 4, Appendix) seem to be quite different from the identified reasoning patterns, making the comparison difficult. I would have liked to see a comparison to system instructions that explicitly tell the model to use one of the reasoning patterns. Furthermore, the results in Table 2 show how susceptible the method is to the scaling factor $\beta$, yet the paper does not provide results for the baselines when similarly scaling the importance sampling ratio (i.e. for natural language guidance). This would be important to ablate the effect of the guidance from the changes to the optimisation objective. Finally, results are limited to Qwen, which is known to already exhibit common reasoning patterns (Gandhi et al., 2025). How does the approach perform on a model that doesn't show these patterns in the first place? Is the guidance still effective, or do the importance sampling weights explode, rendering optimisation difficult in practice?

---

Gandhi et al., 2025. Cognitive Behaviors that Enable Self-Improving Reasoners, or, Four Habits of Highly Effective STaRs. *CoLM*.

Wang et al., 2025. VL-Rethinker: Incentivizing Self-Reflection of Vision-Language Models with Reinforcement Learning. *NeurIPS*.

Deng et al., 2025. OpenVLThinker: Complex Vision-Language Reasoning via Iterative SFT-RL Cycles. *NeurIPS*.

Zhang et al., 2024. Adaptable Logical Control for Large Language Models. *NeurIPS*.

---

> ### Author Rebuttal · Authors · 2026-03-31
>
> We thank the reviewer for the constructive feedback and insightful questions.
>
> Before addressing the individual points, we would like to clarify what we believe is a key misunderstanding about the role of importance sampling in our method, which appears to underlie several concerns.
>
> From the review:
> **“The paper claims accurate importance sampling as an advantage, but this is not unique to the method; any modified behavior policy can in principle use importance sampling by adjusting the policy gradient, as is standard in RL.”**
>
> This is true only when the modified behavior policy is tractable, which is the main challenge in rollout control. For example, manual trigger insertion can alter behavior, but the resulting policy is generally not available in analytic form, so trajectory probabilities cannot be computed exactly. This makes accurate importance-sampling estimation intractable and potentially biased.
>
> A second challenge is how to inject lexical guidance in a principled way, since simple trigger insertion may work for shallow cases but not for structured reasoning patterns (Figure 3b).
>
> Our contribution is therefore to provide a rollout-control framework that enables both
> 1. exact importance-sampling estimation under a tractable guided policy, and
> 2. principled control of rollouts toward target lexical patterns.
>
>
> ## Weakness
>
> **1. What’s the HMM construction complexity, and how does distribution shift affect a fixed HMM distilled only from the initial policy?**
>
> For HMM construction cost, please refer to our first two responses to *reviewer GGP8*. For the distribution shift, we track the noisy rollout rate throughout training and do not observe it increasing, suggesting the HMM remains effective as the policy evolves. If larger shifts arise, a practical mitigation is to re-distill the HMM.
>
> **2. A baseline to compare against is to manually insert the desired patterns at various points during rollout. Such a comparison would justify the complicated machinery proposed by the paper.**
>
> Manual insertion may be feasible for simple triggers such as “wait,” but is much less suitable for structured behaviors, where both placement and context matter. It also makes the modified behavior policy difficult to characterize probabilistically, preventing accurate importance-sampling estimation.
>
> To test this directly, we implemented an **NL-Injection** baseline using XGrammar in the Qwen3-1.7B setting, enforcing the exact keyphrases in Table 1 during rollout as suggested:
>
> | Method | AIME'24 | AIME'25 | MATH500 | AMC'23 | Minerva | Olympiad | Avg. |
> | :--- | :---: | :---: | :---: | :---: | :---: | :---: | :---: |
> | NL Guidance | 14.16 | 10.00 | 69.00 | 47.65 | 23.52 | 33.82 | 33.02 |
> | Ctrl-R | 15.21 | 10.83 | 71.00 | 49.69 | 23.16 | 33.23 | 33.85 |
> | **NL Injection** | 7.70 | 3.95 | 60.8 | 41.87 | 22.05 | 25.81 | 27.03 |
>
> Randomly enforcing lexical patterns during rollout causes **substantial performance degradation**: structural decoding often inserts the target phrase at inappropriate points, disrupting reasoning and producing noisy trajectories. This supports the need for tractable, context-aware guidance as in Ctrl-R.
>
> **3. Comparison to SFT pre-alignment is shown for VLMs but missing for text-only tasks.**
>
> In the VLM setting, Ctrl-R outperforms SFT pre-alignment despite the latter requiring substantial curated data; extending this comparison to text-only tasks would require large-scale data curation and iterative SFT-RL training, which is beyond the current scope.
>
> **4. The NL-guidance comparison is hard to interpret; a comparison to explicit system instructions would be more appropriate.**
>
> This comparison is already included in the VLM setting: Table 4 (bottom) uses system-level instructions that explicitly encourage the target reasoning pattern.
>
> **5. Baseline with scalign factor $\beta$**
>
> The scaling factor $\beta$ in Ctrl-R is introduced to control noisy guided rollouts; in contrast, NL guidance typically causes the model to ignore the hint rather than produce distorted trajectories (Figure 3), so the same mechanism is not directly applicable.
>
> **6. Results are limited to Qwen with common reasoning patterns.**
>
> Figure 3 shows that Qwen3 does not naturally exhibit some enforced counterfactual patterns before training, yet Ctrl-R still guides the model to explore and later use these behaviors more often at evaluation time, suggesting the method is not limited to patterns already present in the base model.
>
> ## Questions
>
> **Why trajectory-level importance sampling?**
>
> Accurate importance sampling is defined at the trajectory level; token-level ratios are an approximation for small policy shifts, whereas Ctrl-R intentionally induces larger shifts, making that approximation less suitable and empirically less stable.
>
> **HMM cost vs. finetuning cost?**
>
> See **Weakness 1**.
>
> **Effect of HMM distribution shift?**
>
> See **Weakness 1**.
>
> **Behavior on models without these patterns?**
>
> See **Weakness 6**.

---

> > ### Author Rebuttal · Reviewer_YVWr · 2026-04-01
> >
> > Many thanks for the response.
> >
> > > This is true only when the modified behavior policy is tractable, which is the main challenge in rollout control. For example, manual trigger insertion can alter behavior, but the resulting policy is generally not available in analytic form, so trajectory probabilities cannot be computed exactly. This makes accurate importance-sampling estimation intractable and potentially biased.
> >
> > I agree, this was an oversight on my part and significantly strengthens the paper. Thank you for clarifying!
> >
> > > This comparison is already included in the VLM setting: Table 4 (bottom) uses system-level instructions that explicitly encourage the target reasoning pattern.
> >
> > I still think the prompts used for natural language guidance could potentially be improved. For example, instead of explicitly forcing keywords, including instructions in the prompt such as "if you notice you get stuck, try to backtrack previous steps" or "re-verify any solution attempt before producing a final answer" could be an interesting comparison.
> >
> > Overall, the authors have successfully addressed my concerns, and I'm increasing my score. I encourage the authors to include the relevant discussion points in the final version of the paper. Good luck with the submission!

---

### Official Review · Reviewer_eyDx · 2026-03-13

**Soundness:** 3
**Presentation:** 3
**Significance:** 3
**Originality:** 3
**Overall Recommendation:** 5
**Confidence:** 4

**Summary:**

### Overview
This paper proposes Ctrl-R, a reinforcement learning framework that improves reasoning capabilities in large language models by explicitly guiding training toward structured reasoning behaviors (like backtracking, induction, counterfactual reasoning). Standard RL struggles to discover such reasoning trajectories because they are rare in unconstrained sampling.

### Key Idea
The authors treat reasoning behaviors as lexical constraints during generation and guide rollout trajectories toward them using a probabilistic controller. This produces a guided behavior policy that encourages exploration of useful reasoning structures while still allowing principled RL optimization.

### Method
Ctrl-R constructs a behavior policy by combining:
- the model’s policy (next-token distribution)
- a guidance function estimating the likelihood that the trajectory will satisfy a reasoning constraint.

To maintain stable training, the framework:
- computes exact importance-sampling weights for off-policy correction
- introduces power-scaled importance weights (β) to balance exploration and unbiased optimization.

**Compliance With Llm Reviewing Policy:**

Affirmed.

**Key Questions For Authors:**

1. **Generality of reasoning constraints.**
   The method relies on manually specified lexical patterns to represent reasoning structures. How sensitive is Ctrl-R to the choice of these patterns, and how well would the framework generalize to domains where such lexical cues are less explicit (e.g., commonsense reasoning or planning tasks)? A convincing analysis or experiment demonstrating robustness to different constraint definitions would strengthen confidence in the method’s general applicability.

2. **Role of the HMM guidance model.**
   The paper distills an HMM to approximate the language model for tractable control. How sensitive are results to the quality of this approximation (e.g., number of latent states, training procedure)? If the HMM poorly approximates the underlying model distribution, could this bias the guidance signal or degrade learning? Clarifying this would help assess the reliability and scalability of the approach.

3. **Ablation on reasoning structure selection.**
   The experiments focus on a fixed set of reasoning behaviors (e.g., backtracking, induction). Did the authors test the effect of including or excluding specific structures during training? Understanding whether improvements are driven by particular behaviors would clarify how essential each component is and whether the framework benefits from larger or automatically discovered structure sets.

**Limitations:**

No. The paper briefly includes an impact statement but does not meaningfully discuss limitations or potential societal impacts.

**Strengths And Weaknesses:**

## Assessment of the Paper

### Soundness
The paper is generally technically sound. The proposed framework (Ctrl-R) is grounded in standard RL formulations and maintains principled optimization through tractable importance sampling. The derivation of the guided behavior policy and the use of power-scaled importance weights are logically motivated and consistent with off-policy RL theory. Empirical evaluations across multiple mathematical reasoning benchmarks for both language and vision–language models support the claims of improved reasoning performance.

### Presentation
The paper is mostly well structured and clearly organized, with a logical flow from motivation to method and experiments. Figures and tables effectively illustrate the framework and empirical results. However, the exposition of the guidance mechanism (HMM + DFA and constrained decoding) is somewhat dense and may be difficult to follow without familiarity with prior work such as Ctrl-G. Additional intuition and simplified explanations of the behavior policy construction could improve readability. The paper does reasonably position itself with respect to prior approaches like reward shaping and natural language guidance.

### Significance
The work addresses an important problem: improving reasoning abilities in language models through reinforcement learning. Guiding exploration toward structured reasoning patterns is a meaningful direction, particularly for reasoning-intensive tasks such as mathematical problem solving. While the empirical gains are modest, the framework provides a potentially useful paradigm for controlling reasoning trajectories during RL training. If extended or generalized, the approach could influence future work on training reasoning-capable LLMs.

### Originality
The paper offers moderate originality. The core components—constrained decoding, guided exploration, and importance sampling in RL—are not individually new. However, the combination of these elements into a unified framework for structured reasoning control, along with the introduction of power-scaled importance weights, represents a creative integration of existing ideas. The work contributes a new perspective on how reasoning behaviors can be systematically explored and reinforced during RL training.

---

> ### Author Rebuttal · Authors · 2026-03-31
>
> We thank the reviewer for the constructive feedback and insightful questions.
>
> **Presentation: The guidance mechanism (HMM + DFA and constrained decoding) is somewhat dense and may be difficult to follow.**
> Thank you for the feedback. We agree that this part can be presented more clearly. At a high level, our key idea is to use a tractable probabilistic model (TPM) to efficiently estimate $P_{\mathrm{TPM}}(\alpha \mid x_t, x_{<t})$ , which lets us bias exploration toward trajectories satisfying the target reasoning constraint while retaining a white-box behavior policy for tractable importance sampling. In the revision, we will improve the exposition by emphasizing this intuition earlier and simplifying the presentation of the HMM+DFA construction.
>
> **Generality of reasoning constraints: How sensitive is Ctrl-R to the choice of these patterns?**
> Ctrl-R is not highly sensitive to the choice of patterns at the level of the learning mechanism. Unlike reward shaping, we do not directly reward the model for producing these patterns; they only modify the exploration distribution during rollout. As a result, suboptimal patterns do not necessarily harm learning. That said, if a pattern consistently leads to poor rollouts or is too infrequent for the distilled HMM to model accurately, it can increase the number of noisy guided trajectories. In practice, these trajectories receive low learning signals via importance sampling, limiting their influence on optimization.
>
> **How sensitive are the results to the quality of this approximation?**
> Approximation quality mainly affects how accurately $P_{\mathrm{HMM}}(\alpha \mid x_t, x_{<t})$ is estimated, and thus how effective the guidance is. Poor approximation increases the fraction of noisy rollouts. Empirically, these usually do not hurt the performance of the reinforcement learning much, since importance sampling downweights them, but they do reduce training efficiency by producing a higher proportion of trajectories with little learning signal.
>
> To improve approximation quality, we make two changes relative to Ctrl-G. First, we use a smaller HMM with 4,096 latent states instead of 32,768, since we found similar perplexity with better efficiency. Second, rather than distilling a general-purpose HMM, we distill directly on the training distribution by using training-set prefixes to generate distillation data. This yields a more accurate approximation of the rollout distribution and improves guidance effectiveness.
>
> **Ablation on reasoning structure selection.**
> We agree that this is an important direction. In this paper, however, our goal is not to identify the best reasoning structures but to study whether controllable rollout guidance can improve RL when given plausible reasoning patterns. We therefore use a small set of hand-designed structures as an initial testbed.
>
> In preliminary experiments, we found that some patterns, such as overthinking awareness and backwarding, were not helpful when trained individually. We nevertheless keep them because Ctrl-R does not force the model to use any specific pattern; it only shifts the exploration distribution, while importance sampling controls how much each trajectory contributes to learning. As a result, unhelpful patterns do not substantially hurt performance as long as useful exploration remains possible. A promising future direction is to select reasoning structures adaptively for each input.

---

> > ### Author Rebuttal · Reviewer_eyDx · 2026-03-31
> >
> > Thank you for answering my questions. I will keep the score. I hope you can open source the implementation soon! Good luck with your reserach and submission!

---

### Decision · Program_Chairs · 2026-04-30

**Decision:**

Accept (spotlight)

**Comment:**

This paper proposes Ctrl-R, a reinforcement learning framework that improves reasoning in large language models by guiding rollout exploration toward structured reasoning patterns (e.g., backtracking, induction, counterfactual reasoning). Reasoning behaviors are encoded as lexical constraints via deterministic finite automata (DFAs), and a tractable probabilistic model (HMM) is distilled to steer trajectory generation. The resulting guided behavior policy admits exact importance-sampling estimation, enabling principled on-policy optimization. A power-scaling factor on importance weights further stabilizes learning from out-of-distribution trajectories. The method is evaluated on mathematical reasoning benchmarks across both language and vision-language models.

The topic is timely and the problem well-motivated: standard RL rarely discovers complex reasoning trajectories under sparse rewards, and Ctrl-R addresses this with a principled exploration framework. The combination of constrained decoding, tractable importance sampling, and power-scaled IS weights is technically sound and novel. Importantly, the guided behavior policy remains analytically tractable, enabling exact importance-sampling estimation. Experiments are conducted with meaningful ablations and behavior analyses.

Multiple reviewers found the exposition dense, e.g. HMM+DFA construction. The authors committed to improving this in the revision. Reviewers also raised concerns about runtime cost. The authors provided empirical benchmarks showing negligible HMM overhead. Reviewer consensus moved firmly toward acceptance following the rebuttal, with all major concerns resolved. The remaining limitations (generalization beyond math, manual constraint specification, SFT comparison for text-only tasks) are acknowledged and appropriate for future work.